# Estimating the Density Ratio between Distributions with High Discrepancy using Multinomial Logistic Regression

**Akash Srivastava**[*]                                                          *akashsri@mit.edu*
*MIT-IBM Watson AI Lab and IBM Research*

**Seungwook Han**[*]                                                              *swhan@mit.edu*
*MIT*

**Kai Xu**                                                                       *xuk@amazon.com*
*Amazon*

**Benjamin Rhodes**                                                          *ben.rhodes@ed.ac.uk*
*School of Informatics, University of Edinburgh*

**Michael U. Gutmann**[*]                                              *michael.gutmann@ed.ac.uk*
*School of Informatics, University of Edinburgh*

**Reviewed on OpenReview:** *https://openreview.net/forum?id=jM8nzUzBWr*

## Abstract

Functions of the ratio of the densities $p/q$ are widely used in machine learning to quantify the discrepancy between the two distributions $p$ and $q$. For high-dimensional distributions, binary classification-based density ratio estimators have shown great promise. However, when densities are *well separated*, estimating the density ratio with a binary classifier is challenging. In this work, we show that the state-of-the-art density ratio estimators perform poorly on *well separated* cases and demonstrate that this is due to distribution shifts between training and evaluation time. We present an alternative method that leverages multi-class classification for density ratio estimation and does not suffer from distribution shift issues. The method uses a set of auxiliary densities $\{m_k\}_{k=1}^K$ and trains a multi-class logistic regression to classify the samples from $p, q$ and $\{m_k\}_{k=1}^K$ into $K + 2$ classes. We show that if these auxiliary densities are constructed such that they overlap with $p$ and $q$, then a multi-class logistic regression allows for estimating $\log p/q$ on the domain of any of the $K + 2$ distributions and resolves the distribution shift problems of the current state-of-the-art methods. We compare our method to state-of-the-art density ratio estimators on both synthetic and real datasets and demonstrate its superior performance on the tasks of density ratio estimation, mutual information estimation, and representation learning. Code: https://www.blackswhan.com/mdre/

## 1 Introduction

Quantification of the discrepancy between two distributions underpins a large number of machine learning techniques. For instance, distribution discrepancy measures known as $f$-divergences (Csiszár, 1964), which are defined as expectations of convex functions of the ratio of two densities, are ubiquitous in many domains of supervised and unsupervised machine learning. Hence, density ratio estimation is often a central task in generative modeling, mutual information and divergence estimation, as well as representation learning (Sugiyama et al., 2012; Gutmann & Hyvärinen, 2010; Goodfellow et al., 2014; Nowozin et al., 2016; Srivastava et al., 2017; Belghazi et al., 2018; Oord et al., 2018; Srivastava et al., 2020). However, in most problems

---

[*]equal contribution

of interest, estimating the density ratio by modeling each of the densities separately is significantly more challenging than directly estimating their ratio for high dimensional densities (Sugiyama et al., 2012). Hence, direct density ratio estimators are often employed in practice.

One of the most commonly used density ratio estimators (DRE) utilizes binary classification via logistic regression (BDRE). Once trained to discriminate between the samples from the two densities, BDREs have been shown to estimate the ground truth density ratio between the two densities (e.g. Gutmann & Hyvärinen, 2010; Gutmann & Hirayama, 2011; Sugiyama et al., 2012; Menon & Ong, 2016). BDREs have been tremendously successful in problems involving the minimization of the density-ratio based estimators of discrepancy between the data and the model distributions even in high-dimensional settings (Nowozin et al., 2016; Radford et al., 2015). However, they do not fare as well when applied to the task of estimating the discrepancy between two distributions *that are far apart or easily separable from each other*. This issue has been characterized recently as the *density-chasm problem* by Rhodes et al. (2020). We demonstrate this in Figure 1 where we employ a BDRE to estimate the density ratio between two 1-D distributions, $p = \mathcal{N}(-1, 0.1)$ and $q = \mathcal{N}(1, 0.2)$ shown in panel (a). Since $p$ and $q$ are considerably far apart from each other, solving the classification problem is relatively simple as illustrated by the visualization of the decision boundary of the BDRE. However, as shown in panel (b), even in this simple setup, BDRE completely fails to estimate the ratio. Kato & Teshima (2021) have also confirmed that most DREs, especially those implemented with deep neural networks, tend to overfit to the training data in some way when faced with the density-chasm problem. Since BDRE-based plug-in estimators are often used in many high-dimensional tasks such as mutual information estimation, representation learning, energy-based modeling, co-variate-shift resolution, and importance sampling (Rhodes et al., 2020; Choi et al., 2021b;a; Sugiyama et al., 2012), resolving density-chasm is an important problem of high practical relevance.

A recently introduced solution to the density-chasm problem, telescopic density-ratio estimation (TRE; Rhodes et al., 2020), tackles it by replacing the easier-to-classify, original logistic regression problem, by a *set* of harder-to-classify logistic regression problems. In short, TRE constructs a set of $K$ auxiliary distributions ($\{m_k\}_{k=1}^K$) to bridge the two target distributions ($p =: m_0$ and $q =: m_{K+1}$) of interest and then trains a set of $K + 1$ BDREs on every pair of *consecutive distributions* ($m_{k-1}$ and $m_k$ for $k = 1, \ldots, K$), which are assumed to be close enough (i.e. not easily separable) for BDREs to work well. After that, an overall density ratio estimate is obtained by taking the cumulative (telescopic) product of all individual estimates.

In this work, we argue that the aforementioned solution to the density chasm problem has an inherent issue of *distribution shift* that can lead to significant inaccuracies in the final density ratio estimation. Notice that the $i$-th BDRE in the chain of BDREs that TRE constructs is only trained on the samples from distributions $m_i$ and $m_{i+1}$. However, post-training, it is typically evaluated on regions where the distributions from the original density ratio estimation problem (i.e. $p$ and $q$) have non-negligible mass. If the high-probability regions of $p$, $q$ and the auxiliary distributions $m_i$ do not overlap, the training and evaluation distributions for the $i$-th BDRE are different. Because of this distribution shift between training and evaluation, the overall density ratio estimation can end up being inaccurate (see Figure 2 and Section 2.1 for further details). We here provide another solution to the density-chasm problem that avoids this distribution shift.

We present MULTINOMIAL LOGISTIC REGRESSION BASED DENSITY RATIO ESTIMATOR (MDRE), a novel method for density ratio estimation that solves the density-chasm problem without suffering from distribution shift. This is done by using auxiliary distributions and *multi-class classification*. MDRE replaces the easy binary classification problem with a *single* harder multi-class classification problem. MDRE first constructs a set of $K$ auxiliary distributions $\{m_k\}_{k=1}^K$ that overlap with $p$ and $q$ and then uses multi-class logistic regression on the $K + 2$ distributions to obtain a density ratio estimator of $\log p/q$. We will show that the multi-class classification formulation avoids the distribution shift issue of TRE.

The key contributions of this work are as follows:

1. We study the state-of-the-art solution to the density-chasm problem (TRE; Rhodes et al., 2020) and identify its limitations arising from distribution shift. We illustrate that this inherent issue can significantly degrade its density ratio estimation performance.

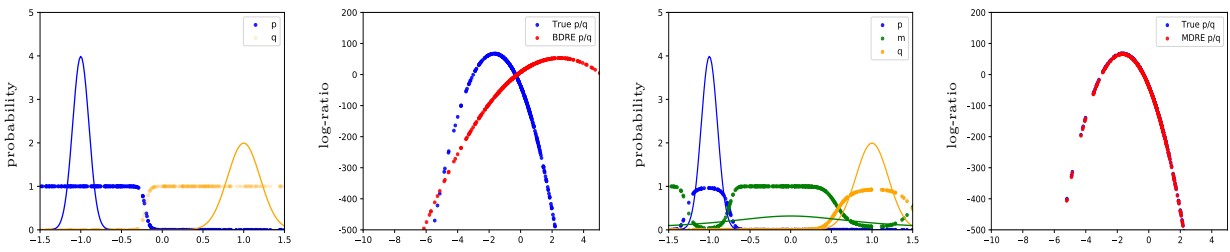

(a) BDRE class probability    (b) BDRE log density-ratio    (c) MDRE class probability    (d) MDRE log density-ratio

Figure 1: BDRE vs proposed MDRE on estimation of log density ratio where $p = \mathcal{N}(-1, 0.1)$ and $q = \mathcal{N}(1, 0.2)$. For MDRE, the auxillary distribution $m$ is Cauchy $\mathcal{C}(0, 1)$. Plots (a) and (c) show the class probabilities $P(Y|x)$ learned for BDRE and MDRE respectively overlayed on the plots of $p$, $q$ and $m$. Plots (b) and (d) show the estimated log density-ratio by BDRE and MDRE respectively. Using auxiliary distribution $m$ allows MDRE to better estimate the log density-ratio.

2. We formally establish the link between multinomial logistic regression and density ratio estimation and propose a novel method (MDRE) that uses auxiliary distributions to train a multi-class classifier for density ratio estimation. MDRE resolves the aforementioned distribution shift issue by construction and effectively tackles the density chasm problem.

3. We construct a comprehensive evaluation protocol that significantly extends on benchmarks used in prior works. We conduct a systematic empirical evaluation of the proposed approach and demonstrate the superior performance of our method on a number of synthetic and real datasets. Our results show that MDRE is often markedly better than the current state-of-the-art of density ratio estimation on tasks such as $f$-divergence estimation, mutual information estimation, and representation learning in high-dimensional settings.

## 2 Related Work

Telescopic density-ratio estimation (TRE, Rhodes et al., 2020) uses a two step, divide-and-conquer strategy to tackle the density-chasm problem. In the first step, they construct $K$ *waymark* distributions $\{m_k\}_{k=1}^K$ by gradually transporting samples from $p$ towards samples from $q$. Then, they train $K$ BDREs, one for each consecutive pair of distributions. This allows for estimating the ratio $r_{p/q}$ as the product of $K + 1$ BDREs, $r_{p/q} := \frac{p}{q} = \frac{p}{m_1} \times \cdots \times \frac{m_K}{q}$. Rhodes et al. (2020) introduced two schemes for creating waymark distributions that ensure that consecutive pairs of distributions are packed *closely enough* so that none of the $K + 1$ BDREs suffer from the density-chasm issue. Hence, TRE addresses the density-chasm issue by replacing the ratio between $p$ and $q$ with a product of $K + 1$ intermediate density ratios that, by design of the waymark distribution, should not suffer from the density-chasm problem. In a new work, Choi et al. (2021b) introduced DRE-$\infty$, a method that takes the number of waymark distributions in TRE to infinity and derives a limiting objective that leads to a more scalable version of TRE.

F-DRE is other interesting related work that comes from Choi et al. (2021a). F-DRE uses a FLOW-based model (Rezende & Mohamed, 2015) which is trained to project samples from a mixture of the two distributions onto a standard Gaussian. They then train a BDRE. It is easy to show that any bijective map will preserve the original density ratio $r_{p/q}$ in the feature space as the Jacobian correction term simply cancels out. However, due to the bijectivity of the FLOW map, such a method cannot bring the projected distributions any closer than the discrepancy between the original distributions. At best, the method can shift the discrepancy between the original distributions along different moments after projection. Due to this issue, we found that F-DRE did not work well for the problems we considered (see experimental results in Section 4). Recently, Liu et al. (2021) introduced an optimization-based solution to the density-chasm problem in exponential family distributions by using (a) normalized gradient descent and (b) replacing the logistic loss with an exponential loss. Finally, while BDRE remains the dominant method of density ratio

estimation in recent literature, prior works, such as Bickel et al. (2008) and Nock et al. (2016), have studied multi-class classifier-based density ratio estimation for estimating ratios between a set of densities against a common reference distribution and its applications in multi-task learning.

## 2.1 TRE's performance can degrade due to training-evaluation distribution shifts

In supervised learning, distribution shift (Quiñonero-Candela et al., 2009) occurs when the training data $(x, y) \sim p_{\text{train}}$ and the test data $(x, y) \sim p_{\text{test}}$ come from two different distributions, i.e. $p_{\text{train}} \neq p_{\text{test}}$. Common training methods, such as those used in BDRE, only guarantee that the model performs well on unseen data that comes from the same distribution as $p_{\text{train}}$. Thus, in the case of distribution shift at test time, the model's performance degrades proportionately to the shift. We now show that a similar distribution shift can occur in TRE when distributions $p$ and $q$ are sufficiently different. Recall that in TRE, we use BDREs to estimate $K + 1$ density ratios $p/m_1, m_2/m_1, \ldots, m_K/q$ that are combined in a telescopic product to form the overall ratio $p/q$. Let us denote the estimates of the $K + 1$ ratios by $\hat{\eta}_1, \ldots, \hat{\eta}_{K+1}$.

Given the theoretical properties of BDRE, for any $i \in \{1, \ldots, K+1\}$, $\hat{\eta}_i$ estimates $r_{m_{i-1}/m_i}$ over the support of $m_i$ (Sugiyama et al., 2012; Gutmann & Hyvärinen, 2010; Menon & Ong, 2016). However, in TRE, when we evaluate the target ratio $p/q$ on the supports of $p$ and $q$, we evaluate the individual $\hat{\eta}_i$ on domains for which we lack guarantees that they perform well. Since the overall estimator for $p/q \approx \hat{\eta}_1 * \cdots * \hat{\eta}_{K+1}$ combines multiple ratio estimators, it suffers from the distribution shift issue if *any* of the individual estimators' performance deteriorates. Thus, if the supports of $\{m_i\}_{i=1}^K$, $p$, and $q$ are different, or when the samples from $\{m_i\}_{i=1}^K$, $p$, and $q$ do not overlap well enough, the training and evaluation domains of the $\hat{\eta}_i$ are different and we expect the ratio estimate $\hat{\eta}_i$ and, in turn, the overall estimator for $p/q$ to be poor. We now illustrate this with a toy example.

We consider estimating the density ratio between $p = \mathcal{N}(-1, 0.1)$ and $q = \mathcal{N}(1, 0.2)$. Since, $p$ and $q$ are well separated, we introduce three auxiliary distributions $m_1, m_2, m_3$ to bridge them, providing the waymarks that TRE needs. The auxiliary distributions $m_1, m_2, m_3$ are constructed with the *linear-mixing* strategy that will be described in Section 3.2. This setup is shown in the top-left panel of Figure 2. We train 4 BDREs $\hat{\eta}_1, \hat{\eta}_2, \hat{\eta}_3, \hat{\eta}_4$ to estimate ratios $p/m_1, m_1/m_2, m_2/m_3$ and $m_3/q$ respectively. We begin by showing that each of the trained BDREs estimates their corresponding density ratio accurately on their corresponding training distributions. To show this, in panels 2-5 in the first row of Figure 2, we evaluate $\hat{\eta}_1, \hat{\eta}_2, \hat{\eta}_3, \hat{\eta}_4$ on samples from their respective denominator densities $m_1, m_2, m_3, q$ and plot them via a scatter plot where the x-axis is labeled with the distribution that we draw the samples from and the y-axis is the log-density ratio (red). We plot the true density ratio in blue for comparison. As evident, red and blue scatter plots overlap significantly, indicating the individual ratio estimators are accurate on their respective denominator (training) distributions.

Next, we evaluate the BDREs $\hat{\eta}_1, \hat{\eta}_2, \hat{\eta}_3, \hat{\eta}_4$, on samples from $p$ and $q$ instead of their corresponding training distributions as before. Distributions $p$ and $q$ are shown in panel 1 of the second row in Figure 2. In the rest of the panels (2-5) in the second row, estimators $\hat{\eta}_1, \hat{\eta}_2, \hat{\eta}_3, \hat{\eta}_4$ are compared to the ground-truth log-density ratios (blue) $p/m_1, m_1/m_2, m_2/m_3$ and $m_3/q$ that are also evaluated on samples from $p$ and $q$. Unlike in row 1, the estimated log-density ratios do not match the ground-truth. This reflects the training-evaluation distribution-shift issues pointed out above. We show now that this deterioration in accuracy on the level of the individual BDREs results in an deterioration of the overall performance of TRE. To this end, we first recover the TRE estimator by chaining the individually trained BDREs via a telescoping product, i.e. $\hat{\eta}_1 * \hat{\eta}_2 * \hat{\eta}_3 * \hat{\eta}_4$ and then evaluate it on samples from all the 5 distributions $p, m_1, m_2, m_3, q$. The results are shown in panels 1-5 of the third row. The estimated log-density ratios (red) do not match the corresponding ground-truth log-density ratios (blue), which demonstrates that the distribution-shift in the training and evaluation distributions of the individual BDREs significantly degrades the overall estimation accuracy of TRE. Additional issues occur when both $p$ and $q$ do not have full support, as discussed in Appendix G.

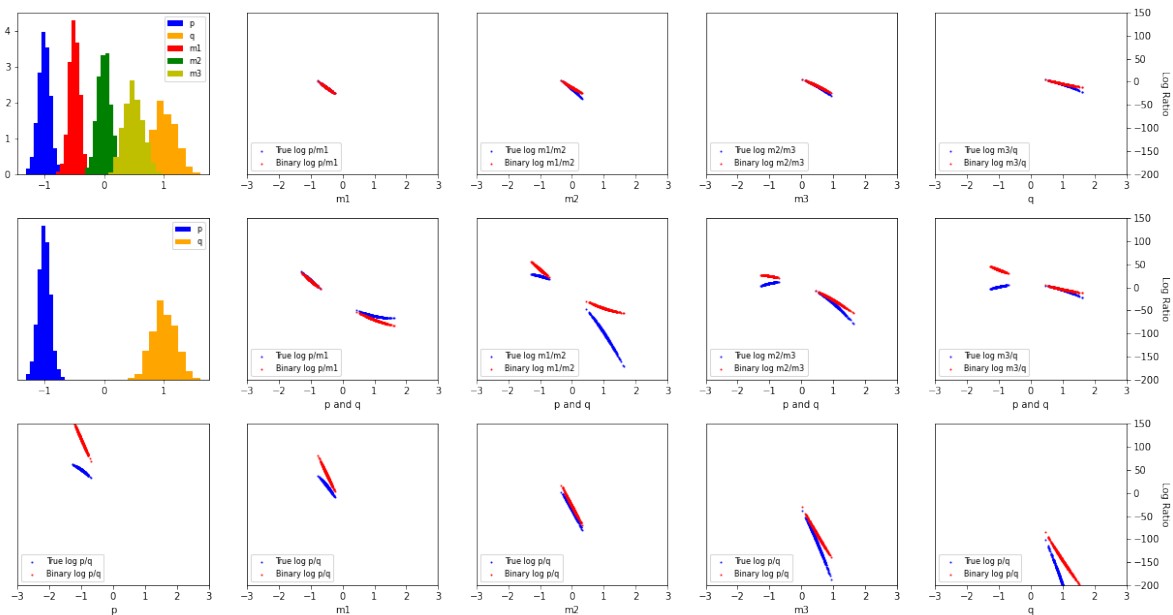

Figure 2: TRE for $p = \mathcal{N}(-1, 0.1)$ and $q = \mathcal{N}(1, 0.2)$ from Figure 1. In all scatter plots, x-axis denotes the sampling distribution and y-axis denotes the log-density-ratio. The density plot in the first row shows $p$, $q$ and the 3 waymarks; the density plot in the second row shows $p$ and $q$ only. The scatter plots in the first row show individual density ratio estimators evaluated on samples from their corresponding training data (denominator density), demonstrating accurate estimation on the training set. The scatter plots in the second show individual density ratio estimators evaluated on samples from $p$ and $q$. The estimation accuracy has degraded notably due to the train-eval distribution shift. The last row shows the performance of the overall density ratio estimator on samples from $p, m_1, m_2, m_3, q$. We see that the overall ratio estimate is significantly affected by the deterioration of the individual ratio estimates, illustrating the sensitivity of TRE to distribution shift problems in case of well-separated distributions.

## 3 Density Ratio Estimation using Multinominal Logistic Regression

We propose Multinomial Logistic Regression based Density Ratio Estimator (MDRE) to tackle the density-chasm problem while avoiding the distribution shift issues of TRE. As in TRE, we introduce a set of $K \geq 1$ auxiliary distributions $\{m_k\}_{k=1}^K$. But, in constrast to TRE, we then formulate the problem of estimating $\log p/q$ as a multi-class classification problem rather than a sequence of binary classification problems. We show that this change leads to an estimator that is accurate on the domain of all $K + 2$ distributions and, therefore, does not suffer from distribution shift.

### 3.1 Loss function

We here establish a formal link between density ratio estimation and multinomial logistic regression. Consider a set of $C$ distributions $\{p_c\}_{c=1}^C$ and let $p_x(x) = \sum_{c=1}^C \pi_c p_c(x)$ be their mixture distribution, with prior class probabilities $\pi_c$.[1] The multi-class classification problem then consists of predicting the correct class $Y \in \{1, \ldots, C\}$ from a sample from the mixture $p_x$. For this purpose, we consider the model

$$P(Y = c|x; \theta) = \frac{\pi_c \exp(h_\theta^c(x))}{\sum_{k=1}^C \pi_k \exp(h_\theta^k(x))}, \tag{1}$$

---

[1]In our simulations, we will use a uniform prior over the classes.

where the $h_\theta^c(x)$, $c = 1, \ldots, C$ are unnormalized log probabilities parametrized by $\theta$. We estimate $\theta$ by minimizing the negative multinomial log-likelihood (i.e. the softmax cross-entropy loss) $\mathcal{L}(\theta)$

$$\mathcal{L}(\theta) = -\sum_{c=1}^{C} \pi_c \mathbb{E}_{x \sim p_c}[\log P(Y = c|x; \theta)] = \sum_{c=1}^{C} \pi_c \mathbb{E}_{x \sim p_c}\left[ - \log \pi_c - h_\theta^c + \log \sum_{k=1}^{C} \pi_k \exp(h_\theta^k(x)) \right], \quad (2)$$

where, in practice, the expectations are replaced with a sample average. We denote the optimal parameters by $\theta^* = \arg \min_\theta \mathcal{L}(\theta)$. To ease the theoretical derivation, we consider the case where the $h_\theta^c(x)$ are parametrized in such a flexible way that we can consider the above loss function to be a functional of $C$ functions $h_1, \ldots, h_C$,

$$\mathcal{L}(h_1, \ldots, h_C) = \sum_{c=1}^{C} \pi_c \mathbb{E}_{x \sim p_c}\left[ - \log \pi_c - h_c + \log \sum_{k=1}^{C} \pi_k \exp(h_k(x)) \right]. \quad (3)$$

The following propositions shows that minimizing $\mathcal{L}(h_1, \ldots, h_C)$ allows us to estimate the log ratios between any pair of the $C$ distributions $p_c$.

**Proposition 3.1.** *Let $\hat{h}_1, \ldots, \hat{h}_C$ be the minimizers of $\mathcal{L}(h_1, \ldots, h_C)$ in equation 3. Then the density ratio between $p_i(x)$ and $p_j(x)$ for any $i, j \leq C$ is given by*

$$\log \frac{p_i(x)}{p_j(x)} = \hat{h}_i(x) - \hat{h}_j(x) \quad (4)$$

*for all $x$ where $p_x(x) = \sum_c \pi_c p_c(x) > 0$.*

*Proof.* We first note that the sum of expectations $\sum_{c=1}^{C} \pi_c \mathbb{E}_{x \sim p_c}$ in equation 3 is equivalent to the expectation with respect to the mixture distribution $p_x$. Writing the expectation as an integral we obtain

$$\mathcal{L}(h_1, \ldots, h_C) = \sum_{c=1}^{C} \pi_c \mathbb{E}_{x \sim p_c}[- \log \pi_c - h_c(x)] + \int p_x(x)[\log \sum_{k=1}^{C} \pi_k \exp(h_k(x))]dx. \quad (5)$$

The functional derivative of $\mathcal{L}(h_1, \ldots, h_C)$ with respect to $h_i$, i=1..., C, equals

$$\frac{\delta \mathcal{L}}{\delta h_i} = -\pi_i p_i(x) + p_x(x) \frac{\pi_i \exp(h_i(x))}{\sum_{k=1}^{C} \pi_k \exp(h_k(x))} \quad (6)$$

for all $x$ where $p_x(x) > 0$. Setting the derivative to zero gives the necessary condition for an optimum

$$\frac{\pi_i p_i(x)}{p_x(x)} = \frac{\pi_i \exp(h_i(x))}{\sum_{k=1}^{C} \pi_k \exp(h_k(x))}, \qquad i = 1, \ldots, C, \text{ and for all } x \text{ where } p_x(x) > 0. \quad (7)$$

The left-hand side of equation 7 equals the true conditional probability $P^*(Y = i|x) = \frac{\pi_i p_i(x)}{p_x(x)}$. Hence, at the critical point, $\hat{h}_1, \ldots, \hat{h}_C$ are such that $P^*(Y|X)$ is correctly estimated. From equation 7, it follows that for two arbitrary $i$ and $j$, we have $(\pi_i p_i)/(\pi_j p_j) = (\pi_i \exp(\hat{h}_i))/(\pi_j \exp(\hat{h}_j))$ i.e.

$$\log \frac{p_i(x)}{p_j(x)} = \hat{h}_i(x) - \hat{h}_j(x) \quad (8)$$

for all $x$ where $p_x(x) > 0$, which concludes the proof. $\square$

*Remark* 3.2 (Identifiability). While we have $C$ unknowns $h_1, \ldots, h_C$ and $C$ equations in equation 7, there is a redundancy in the equations because

$$\sum_{i=1}^{C} \frac{\pi_i p_i(x)}{p_x(x)} = \sum_{i=1}^{C} \frac{\pi_i \exp(h_i(x))}{\sum_{k=1}^{C} \pi_k \exp(h_k(x))} = \frac{\sum_{i=1}^{C} \pi_i \exp(h_i(x))}{\sum_{k=1}^{C} \pi_k \exp(h_k(x))} = 1$$

This means that we cannot uniquely identify all $h_i$ by minimising equation 3. However, the difference $h_i - h_j$, for $i \neq j$, can be identified and is equal to the desired log ratio between $p_i$ and $p_j$ per equation 8.

*Remark* 3.3 (Effect of parametrisation and finite sample size). In practice, we only have a finite amount of training data and the parametrisation introduces constraints on the flexibility of the model. With additional assumptions, e.g. that the true density ratio $\log p_i(x) - \log p_j(x)$ can be modeled by the difference of $h_\theta^i(x)$ and $h_\theta^j(x)$, we show in Appendix A that our ratio estimator is consistent. We here do not dive further into the asymptotic properties of the estimator but focus on the practical applications of the key result in equation 8.

Importantly, equation 8 allows us to estimate $r_{p/q}$ by formulating our ratio estimation problem as a multinomial nonlinear regression problem as summarized in the following corollary.

**Corollary 3.4.** *Let the distributions of the first two classes be $p$ and $q$, respectively, i.e. $p_1 \equiv p, p_2 \equiv q$, and the remaining $K$ distributions be equal to the auxiliary distributions $m_i$, i.e. $p_3 \equiv m_1, \ldots, p_{K+2} \equiv m_K$. Then*

$$\log \hat{r}_{p/q}(x) = \hat{h}_1(x) - \hat{h}_2(x). \tag{9}$$

*Remark* 3.5 (Free from distribution shift issues). Since equation 8 holds for all $x$ where the mixture $p_x(x) > 0$, the estimator $\hat{r}_{p/q}(x)$ in equation 9 is valid for all $x$ in the union of the domain of $p, q, m_1, \ldots, m_K$. This means that MDRE does not suffer from the distribution shift problems that occur when solving a sequence of binomial logistic regression problems as in TRE. We exemplify this in Section 3.3 after introducing three schemes to construct the auxiliary distributions $m_1, \ldots, m_K$.

## 3.2 Constructing the auxiliary distributions

In MDRE, auxiliary distributions need to be constructed such that they have overlapping support with the empirical densities of $p$ and $q$. This allows the multi-class classification probabilities to be better calibrated and leads to an accurate density ratio estimation. We demonstrate this in panel (c) of Figure 1, where $p = \mathcal{N}(-1, 0.1)$ and $q = \mathcal{N}(1, 0.2)$ and the single auxiliary distribution $m$ is set to be Cauchy $\mathcal{C}(0, 1)$ that clearly overlaps with the other two distributions. The classification probabilities are shown as the scatter plot that is overlayed on the empirical densities of these distributions. Compared to the BDRE case in panel (a), which has high confidence in regions without any data, the multi-class classifier assigns, for $p$ and $q$, high class probabilities only over the support of the data and not where there are barely any data points from these two distributions. Moreover, the auxiliary distribution well covers the space where $p$ and $q$ have low density, which provides the necessary training data to inform the values of $\hat{h}_1(x)$ and $\hat{h}_2(x)$ in that area, which leads to an accurate estimate of the log-density ratio shown in panel (d). This is contrast to BDRE in panel (a) where the classifier, while constrained enough to get the classification right, is not learned well enough to also get the density ratio right (panel b). This subtle, yet important distinction between the usage of auxiliary distributions in MDRE compared to BDRE and TRE enables MDRE to generalize on out-of-domain samples, as we will demonstrate in Section 4.1.

Next, we briefly describe three schemes to construct auxiliary distributions for MDRE and leave the details to Appendix B: **(1) Overlapping Distribution** Unlike TRE, the formulation of MDRE does not require "gradually bridging" the two distributions $p$ and $q$, hence, we introduce a novel approach to constructing auxiliary distributions. We define $m_k$ as any distribution whose samples overlap with both $p$ and $q$, and $p << m_k, q << m_k$. This includes heavy-tailed distributions (e.g. Cauchy, Student-t), normal distributions, uniform distributions, or their mixtures. We use this scheme in all low-dimensional simulations. **(2) Linear Mixing** In this scheme, $m_k$ is defined as the distribution of the samples generated by linearly combining samples $X_p = \{x_p^i\}_{i=1}^N$ and $X_q = \{x_q^i\}_{i=1}^N$ from distributions $p$ and $q$, respectively. The generative process for a single sample $x_{m_k}^i$ from $m_k$ is given by $x_{m_k}^i = (1 - \alpha_k)x_p^i + \alpha_k x_q^i$, with $x_p \in X_p, x_q^i \in X_q$. This construction is similar to the linear combination scheme for auxiliary distributions introduced by Rhodes et al. (2020) with a few key differences that we expand upon in Appendix B. One difference is that $\alpha_k$ is not limited t o $0 < \alpha_k < 1$, which allows for non-convex mixtures that completely surround both $p$ and $q$. We use this construction scheme in higher-dimensional simulations. **(3) Dimension-wise Mixing** This construction scheme was introduced in (Rhodes et al., 2020). Samples from the single auxiliary distribution $m$ are obtained by combining different subsets of dimensions from samples from $p$ and $q$. We use this scheme for experiments involving high-dimensional image data.

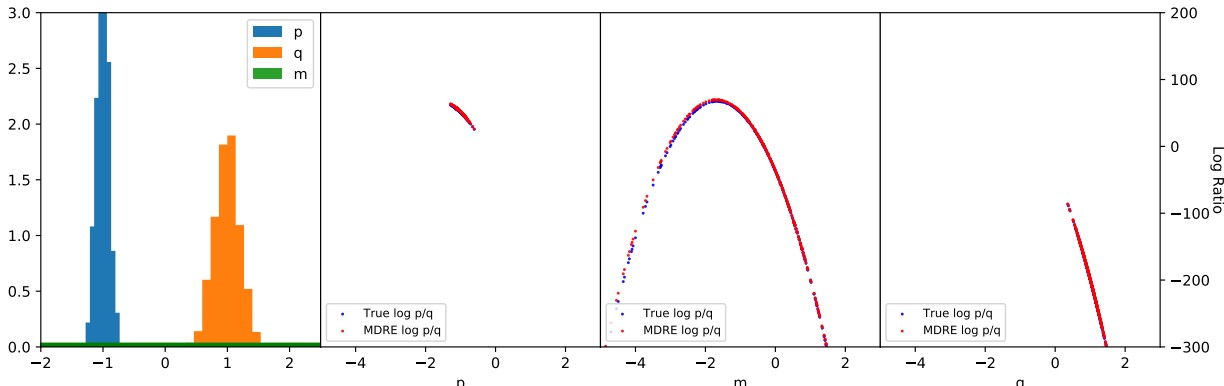

Figure 3: MDRE with $m = \mathcal{C}\mathrm{auchy}(0,1)$. The first density plots shows the target densities as well as the auxiliary distribution (in green, hard to see due to heavy tail and the range of axes). The three scatter plots show the estimated (red) and true (blue) density ratio $p/q$ evaluated on samples from $p, m$, and $q$. MDRE accurately estimates the ratio across the input domain. Contrast with Figure 2.

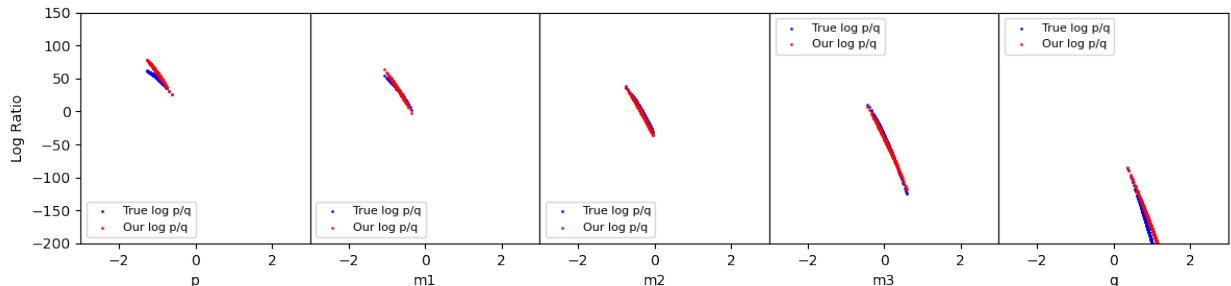

Figure 4: MDRE using TRE's auxiliary distributions. Each scatter plot shows the overall log-density ratio estimates on samples from the distribution on the x-axis (MDRE in red and true ratio in blue). MDRE is capable of accurately estimating ratios on all samples. Contrast with the bottom row of Figure 2.

### 3.3   Free from distribution-shift problems

We continue with the example task of estimating the density ratio between p = N (-1, 0.1) and q = N (1, 0.2) and here illustrate Remark 3.5 that MDRE does not suffer from the distribution shift problem identified in Section 2.1. We test MDRE with two types auxiliary distributions. First using a heavy-tailed distribution ($m = \mathcal{C}\mathrm{auchy}(0,1)$), under the overlapping distributions scheme, and second, with waymark distributions $m_1, m_2, m_3$ as used by TRE in Figure 2 using their linear-mixing construction scheme.

Figure 3 shows the result for the heavy tailed $\mathcal{C}\mathrm{auchy}(0,1)$ auxiliary distribution (green, shown in the left most figure). We can see that the log-ratio learned by MDRE is accurate even beyond the empirical support of $p$ and $q$. This is because MDRE is trained on samples from the mixture of $p, q$ and $m$ and hence, per Remark 3.5, does not encounter distribution-shift, over the support of the mixture distribution. Figure 4 shows the result when using the auxiliary distributions of TRE that we used in Figure 2. We see that the learned log-ratio well matches the true log-ratio on samples from $p$ and $q$, as well as the auxiliary distributions. This can be directly compared to third row of Figure 2 where TRE suffers from distribution shift problems and does not yield a well-estimated log-ratio. Note that we do not present results that correspond to the second row of 2 since the estimation of the log-ratio in MDRE *does not* depend on any intermediate density ratios.

| $p$ | $q$ | True KL | BDRE | TRE | F-DRE | MDRE (ours) |
|---|---|---|---|---|---|---|
| $\mathcal{N}(-1, 0.08)$ | $\mathcal{N}(2, 0.15)$ | 200.27 | $21.74 \pm 4.10$ | $136.05 \pm 5.91$ | $14.87 \pm 1.72$ | $203.32 \pm 2.01$ |
| $\mathcal{N}(-2, 0.08)$ | $\mathcal{N}(2, 0.15)$ | 355.82 | $20.22 \pm 3.64$ | $208.11 \pm 18.31$ | $14.22 \pm 5.30$ | $360.35 \pm 1.37$ |

Table 1: 1D density ratio estimation task for $p$ and $q$ with large first-order and higher-order differences. In all cases, MDRE outperforms all the baselines.

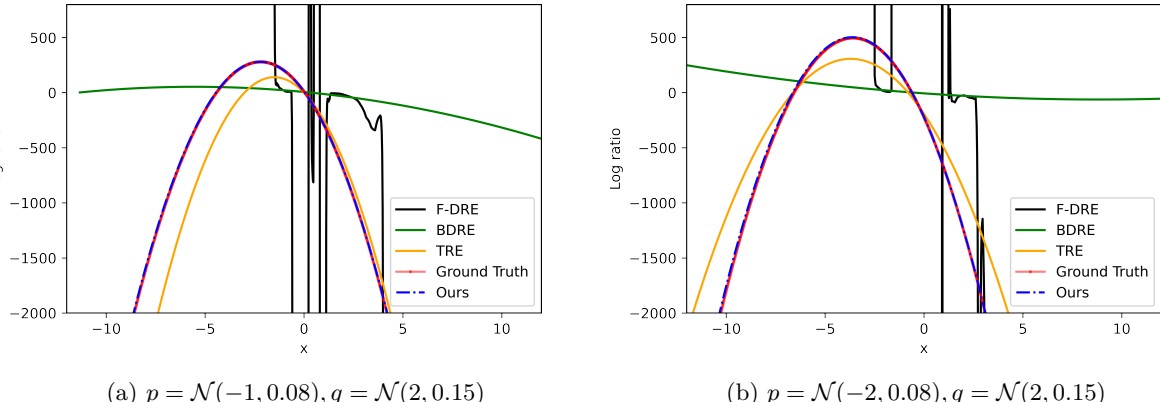

(a) $p = \mathcal{N}(-1, 0.08), q = \mathcal{N}(2, 0.15)$  (b) $p = \mathcal{N}(-2, 0.08), q = \mathcal{N}(2, 0.15)$

Figure 5: Log density-ratio estimates corresponding to the numbers reported in Table 1. Note that the ground truth and MDRE curves are overlapping, while all the other estimators are significantly worse.

## 4 Experiments

We here provide an empirical analysis of MDRE on both synthetic and real data, showing that it performs better than previous methods—BDRE, TRE, and F-DRE—on three different density ratio estimation tasks. We consider cases where numerator and denominator densities differ because their mean is different, i.e. $p$ and $q$ exhibit first-order discrepancies (FOD), and cases where the difference stems from different higher-order moments (higher-order discrepancies, HOD). Ratio estimation is closely related to KL divergence and mutual information estimation since the KL divergence is the expectation of the log-ratios under $p$, and mutual information can be expressed as a KL divergence between joint and product of marginals. Being quantities of core interest in machine learning, we will use them to evaluate the ratio estimation methods.

### 4.1 1D Gaussian experiments with large KL divergence

In the following 1D experiments, we consider two scenarios, one where $p = \mathcal{N}(-1, 0.08)$ and $q = \mathcal{N}(2, 0.15)$, and one where the mean of $p$ is shifted to $-2$ in order to increase the degree of separation between the two distributions. In both cases, MDRE's auxiliary distribution $m$ is $\mathcal{C}\text{auchy}(0,1)$, so that we have a three-class classification problem ($p$, $q$, $m$) and three functions $h_\theta^i$ that parameterize the classifier of MDRE. The three functions are quadratic polynomials of the form $w_1 x^2 + w_2 x + b$. For all the methods we set the total number of samples to 100K.[2] We provide the exact hyperparameter settings for MDRE and other baselines in Table 5 in Appendix C.

Table 1 shows the results. We can see that MDRE yields more accurate estimates of the KL divergences than the baselines, which are off by a significant margin.

We note that KL estimation only requires evaluating the log-ratios on samples from the numerator distribution $p$. In Figure 5, we thus show results for all methods where we evaluate the estimated log-ratios on a wide interval (-12, 12). The figure shows that none of the baseline methods can accurately estimate the ratio well on the whole interval while MDRE performs well overall. This is important be-

---

[2]We found that MDRE's results are unchanged even when using smaller sample sizes of 1K or 10K, see Table 4 in Appendix C.

| Dim | $\mu_1, \mu_2$ | True MI | BDRE | TRE | F-DRE | MDRE (ours) |
|-----|------|---------|------|-----|-------|-------------|
| 40 | 0, 0 | 20 | $10.90 \pm 0.04$ | $14.52 \pm 2.07$ | $14.87 \pm 0.33$ | $18.81 \pm 0.15$ |
| | -1, 1 | 100 | $29.03 \pm 0.09$ | $33.95 \pm 0.14$ | $13.86 \pm 0.26$ | $119.96 \pm 0.94$ |
| 160 | 0, 0 | 40 | $21.47 \pm 2.62$ | $34.09 \pm 0.21$ | $12.89 \pm 0.87$ | $38.71 \pm 0.73$ |
| | -0.5, 0.6 | 136 | $24.88 \pm 8.93$ | $69.27 \pm 0.24$ | $13.74 \pm 0.13$ | $133.64 \pm 3.70$ |
| 320 | 0, 0 | 80 | $23.47 \pm 9.64$ | $72.85 \pm 3.93$ | $9.17 \pm 0.60$ | $87.76 \pm 0.77$ |
| | -0.5, 0.5 | 240 | $24.86 \pm 4.07$ | $100.18 \pm 0.29$ | $10.53 \pm 0.03$ | $217.14 \pm 6.02$ |

Table 2: High-dimensional mutual information estimation task. MDRE is able to accurately estimate the MI often by a very large margins.

cause it means that the ratio is well estimated in regions where $p$ and $q$ have little probability mass. These results demonstrate the effectiveness of MDRE with a single auxiliary distribution whose samples overlaps with those from both $p$ and $q$, in lieu of using a chain of BDREs with up to $K = 28$ closely-packed auxiliary distributions as used by TRE. Please see Appendix C for additional results and details.

To provide further clarity into MDRE's density ratio estimation behavior, we analyze the uncertainty of its log ratio estimates using Bayesian analysis. We use a standard normal prior on the classifier parameters and obtain posterior samples with Hamiltonian Monte-Carlo. These posterior samples then yield samples of the density ratio estimates. Figure 6 shows that the high accuracy of MDRE's KL divergence estimates can be attributed to MDRE being confidently accurate around the union of the high density regions of both $p$ and $q$. A more detailed analysis is provided in Appendix D.

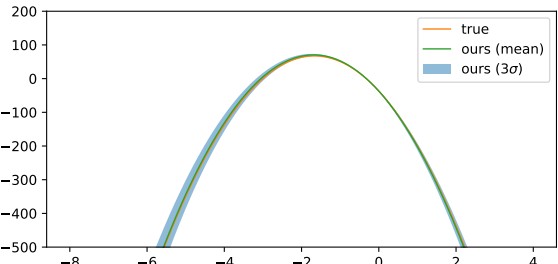

Figure 6: Bayesian analysis of MDRE for $p = \mathcal{N}(-1.0, 0.1), q = \mathcal{N}(1.0, 0.2)$.

## 4.2 High dimensional experiments with large MI

Following Rhodes et al. (2020), we use the MI estimation benchmark from Belghazi et al. (2018); Poole et al. (2019) to evaluate MDRE on a more challenging, higher-dimensional problem. In this task, the goal is to estimate the mutual information between a standard normal distribution and a Gaussian random variable $x \in \mathbb{R}^{2d}$ with a block-diagonal covariance matrix where each block is $2 \times 2$ with ones on the diagonal and $\rho$ on the off-diagonal. The correlation coefficient $\rho$ is computed from the number of dimensions and the target mutual information $I = -d/2 \log(1 - \rho^2)$. Since this problem construction only induces higher-order discrepancies (HOD), we added an additional challenge by moving the means of the two distributions, thus additionally inducing first-order discrepancies (FOD).

For MDRE, we model the $h_\theta^i$ with quadratic functions of the form $x^T W_1 x + W_2 x + b$. We use linear-mixing to construct each $m_k$, where $K = 3$ or $K = 5$. In Appendix E, we provide the exact configurations for MDRE in Table 6 and explain how to choose $m$ and $K$ in practice.

Table 2 shows the results for each MI task averaged across 3 runs with different random seeds. MDRE outperforms all baselines in the original MI task where the means of the distribution are the same. The difference between the performance of MDRE and the baselines is particularly stark when the means are allowed to be nonzero. Only MDRE estimates the MI reasonably well while all baselines dramatically underestimate it. We further note that MDRE only uses up to 5 auxiliary distributions, lowering its compute requirements compared to TRE, which is the next best performing method and uses up to 15 auxiliary distributions for its telescoping chain.

We found that the resolution proposed by Kato & Teshima (2021) to overcome the over-fitting issue in Bregman Divergence minimization-based DREs, does not work well in practice. On the high-dimensional

setup of row 2 in Table 2, while the ground truth MI is 100, and MDRE estimates it as $119 \pm 0, 94$, the best model from Kato & Teshima (2021) yields 1.60, significantly underestimating the true value and being a factor of ten smaller than the classifier-based DRE baselines. For further results, such as plots of estimated log ratio vs. ground-truth log ratio, training curves, and more, please see Appendix E.

Above, following prior work, we evaluated the methods on problems where $p$ and $q$ are normal distributions. To enhance this analysis, we further evaluate MDRE on the three new experimental setups below. The results are summarized in Table 3.

**Breaking Symmetry** In our high-dimensional experiments reported in Table 2, the means of the Gaussian distributions $p$ and $q$ were symmetric around zero in the majority of cases. In order to ensure that this symmetry did not provide an advantage to MDRE, we also evaluate it on Gaussians $p$ and $q$ with randomized means. The results are shown in rows 2 and 6 of Table 3. We see that MDRE continues to estimate the ground truth KL divergence accurately, demonstrating that it did not benefit unfairly from the symmetry of distributions around zero.

**Model Mismatch** In rows 4, 5, 7, and 8 of Table 3, we evaluate MDRE by replacing one or both distributions $p$ and $q$ with a Student-t distribution of the same scale with randomized means. For the Student-t distributions, we set the degrees of freedom as 5, 10 or 20. These experiments test how well MDRE performs when there is model mismatch, i.e. how MDRE performs using the same quadratic model that was used when $p$ and $q$ were set to be Gaussian with lighter tails. We find that MDRE is still able to accurately estimate the ground truth KL in these cases. We found the same to be true for other test distributions such as a Mixture of Gaussians (shown in row 3 of Table 3).

**Finite Support** $p$ **and** $q$ Finally, we test MDRE on another problem where $p$ and $q$ are finite support distributions that have both FOD and HOD. This is done by setting $p$ and $q$ to be truncated normal distributions, as shown in row 1 of Table 3. We also set $m$ to be a truncated normal distribution with its scale set to 2 to allow it to have overlap with both $p$ and $q$. This setting is similar to the 1D Gaussian example illustrated in Section 3.3 and MDRE manages to estimate the ground-truth KL divergence accurately.

| Dim | $p$ | $q$ | $m$ | True KL | Est. KL |
|---|---|---|---|---|---|
| 1 | Truncated Normal loc=-1, scale=0.1 support=(-1.1,-0.9) | Truncated Normal loc=1, scale=0.2 support=(-1.1,1.2) | Truncated Normal loc=-1, scale=2 support=(-1.1,1.2) | 50.65 | 52.35 |
| 160 | Normal loc=R(-.5,.5), cov=$2 \times 2$ BD | Normal loc=R(-.5,.5), cov=$I$ | Linear Mixing | 54.29 | 54.10 |
| 160 | Normal loc=-1, cov=$2 \times 2$ BD | MoG: 0.5*Normal(0.9, $I$) + 0.5*Normal(1.1,$I$) | Linear Mixing | 105.60 | 98.27 |
| 160 | Student T loc=R(-.5,.5), scale=$2 \times 2$ BD, df=5 | Student T loc=R(-.5,.5), scale=I, df=5 | Linear Mixing | 51.26 | 49.01 |
| 320 | Student T loc=R(-.5,.5), scale=$2 \times 2$ BD, df=10 | Student T loc=R(-.5,.5), scale=I, df=10 | Linear Mixing | 53.82 | 51.03 |
| 320 | Normal loc=R(-1,1), cov=$2 \times 2$ BD | Normal loc=R(-1,1), cov=$I$ | Linear Mixing | 110.05 | 102.63 |
| 320 | Student T loc=R(-1,1), scale=$2 \times 2$ BD, df=10 | Student T loc=R(-1,1), scale=$I$, df=10 | Linear Mixing | 103.12 | 113.53 |
| 320 | Normal loc=0, cov=$2 \times 2$ BD | Student T loc=0, scale=$I$, df=20 | Linear Mixing | 82.02 | 83.63 |

Table 3: Robustness evaluation for MDRE. Here R(a,b) stands for randomized mean vector where each dimension is sampled uniformly from the interval $(a, b)$. MDRE is able to consistently estimate the ground-truth KL with high accuracy in all of the cases.

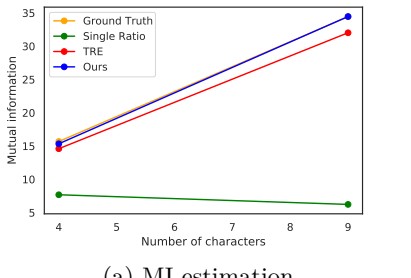 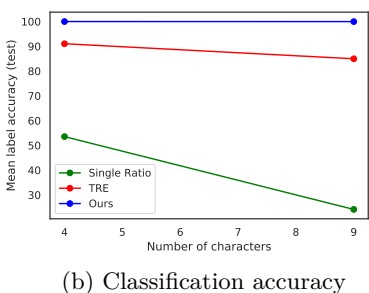 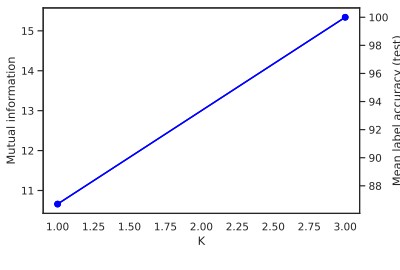

|     |     |     |
| :-: | :-: | :-: |
| (a) MI estimation | (b) Classification accuracy | (c) Varying $K$ (# of auxiliary dist.) |

Figure 7: SpatialMultiOmniglot representation learning results. Plot (a) shows the MI estimated by the three methods, MDRE is able to estimate the ground truth MI very accurately. Plot (b) shows the resulting classification accuracy and plot (c) the impact of varying the number of auxiliary distributions on MI estimation with MDRE.

## 4.3 Representation learning for SpatialMultiOmniglot

In order to benchmark MDRE on large-scale real-world data, following the setup from Rhodes et al. (2020), we apply MDRE to the task of mutual information estimation and representation learning for the Spatial-MultiOmniglot problem (Ozair et al., 2019). The goal is to estimate the mutual information between $u$ and $v$ where $u$ is a $n \times n$ grid of Omniglot characters from different Omniglot alphabets and $v$ is a $n \times n$ grid containing (stochastic) realizations of the next characters of the corresponding characters in $u$. After learning, we evaluate the representations from the encoder with a standard linear evaluation protocol (Oord et al., 2018). For MDRE, similarly to TRE, we utilize a separable architecture commonly used in the MI-based representation learning literature and model the unnormalized log-scores $h_\theta^i$ with functions of the form $g(u)^T W f(v)$ where $g$ and $f$ are 14-layer convolutional ResNets (He et al., 2015). While this model amounts to sharing of parameters across the $h_\theta^i$, we would like to emphasize that in all preceding examples, we did not share parameters among the $h_\theta^i$. We construct the auxiliary distributions via dimension-wise mixing.

We here only compare MDRE to the single ratio baseline and TRE because Rhodes et al. (2020, Figure 4) already demonstrated that TRE significantly outperforms both Contrastive Predictive Coding (CPC) (Oord et al., 2018) and Wasserstein Predictive Coding (WPC) (Ozair et al., 2019) on exactly the same task. Please refer to Appendix F for the detailed experimental setup.

As can be seen in Figure 7a, MDRE performs better than TRE and the single ratio baseline, exactly matching the ground truth MI. This improvement in MI estimation is reflected in the representations. Figure 7b illustrates that MDRE's encoder learns representations that achieve $\sim100\%$ Omniglot character classification for both $d = n^2 = 4, 9$. On the other hand, the performances of the single ratio estimator and TRE (using the same exact dimension-wise mixing to construct auxiliary distributions) both degrade as the complexity of the task increases, with TRE only reaching up to 91% and 85% for $d = 4$ and $d = 9$, respectively. All models were trained with the same encoder architecture to ensure fair comparison.

We further studied the effect of changing $K$ in the $d = 4$ setup. For $K = 1$, we aggregate all the dimension-wise mixed samples into 1 class, whereas for $K = 3$, we separate them into their respective classes (corresponding to the number of dimensions mixed). We illustrate this effect in Figure 7c. In line with the finding of Ma & Collins (2018), increasing the number of K not only helps MDRE to reach the ground truth MI, but also the quality of representations improves from 86.7% to 100% test classification accuracy.

## 5 Discussion

In this work, we presented the multinomial logistic regression based density ratio estimator (MDRE), a new method for density ratio estimation that had better finite sample (non-asymptotic) performance in our simulations than current state-of-the-art methods. We showed that it addresses the sensitivity to possible distribution-shift issues of the recent method by Rhodes et al. (2020). MDRE works by introducing auxiliary

distributions that have overlapping support with the numerator and denominator distributions of the ratio. It then trains a multinomial logistic regression model to estimate the density-ratio. We demonstrated that MDRE is both theoretically grounded and empirically strong, and that it sets a new state of the art for high-dimensional density ratio estimation problems.

However, there are some limitations. First, while the ratio was well estimated in our empirical studies, we do not provide any bounds on the estimation, meaning that estimated KL divergences or mutual information values may be over- or underestimated. Second, the choice of the auxiliary distribution, $m$, is an important factor of consideration that significantly impacts the performance of MDRE. While in this work we demonstrate the efficacy of three schemes for constructing the auxiliary distribution, empirically, it is, by no means, an exhaustive study. We hope to address these issues in future work, including the development of learning-based approaches to auxiliary distribution construction.

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

# A Consistency

In Section 3.1, we focused on properties of the loss function $\mathcal{L}(h_1, \ldots, h_C)$ in equation 3. The arguments of the loss functions were the functions $h_i$, and the loss function was defined in terms of expectations over $p_c$. This simplified the analysis and provided important insights but does not correspond to practical settings. Here, we relax the assumptions: We first consider the loss $\mathcal{L}(\theta)$ in equation 2 where the functions $h_c$ are parameterized by some parameters $\theta$. Then we consider the case where the expectations are replaced by a sample average based on $n$ samples. The corresponding loss function will be denote by $\mathcal{L}_n(\theta)$. The main point of this section is to derive conditions under which minimizing $\mathcal{L}_n(\theta)$ leads to the results in Section 3.1, obtained by minimizing $\mathcal{L}(h_1, \ldots, h_C)$.

**Lemma A.1.** *Denoting the true conditional distribution by $P^*(Y|x)$ we have*

$$\hat{\theta} = \arg\min_{\theta} \mathcal{L}(\theta) = \arg\min_{\theta} \mathbb{E}_{p_x(x)} KL\left(P^*(Y|x) \,||\, P(Y|x; \theta)\right) \tag{10}$$

*where $P(Y|x; \theta)$ is defined in equation 1.*

*Proof.* We start with the definition of $\mathcal{L}(\theta)$ in equation 2:

$$\mathcal{L}(\theta) = -\sum_{c=1}^{C} \pi_c \mathbb{E}_{x \sim p_c}[\log P(Y = c|x; \theta)] \tag{11}$$

The sum of weighted expectations $\sum_{c=1}^{C} \pi_c \mathbb{E}_{x \sim p_c}$ corresponds to a joint expectation over $p(Y, x)$. Decomposing the joint as $p(x)P^*(Y|x)$, we thus obtain

$$\mathcal{L}(\theta) = -\mathbb{E}_{p(x)} \mathbb{E}_{P^*(Y|x)}[\log P(Y|x; \theta)] \tag{12}$$

and

$$\mathcal{L}(\theta) + \mathbb{E}_{p(x)} \mathbb{E}_{P^*(Y|x)} \log P^*(Y|x) = \mathbb{E}_{p(x)} \mathbb{E}_{P^*(Y|x)} \log \frac{P^*(Y|x)}{P(Y|x; \theta)} \tag{13}$$

$$= \mathbb{E}_{p_x(x)} \mathrm{KL}\left(P^*(Y|x) \,||\, P(Y|x; \theta)\right) \tag{14}$$

The claim follows since the added term does not depend on $\theta$. $\qquad\square$

If the true conditional $P^*(Y|x)$ is part of the parametric family $\{P(Y|x; \theta)\}_\theta$, $\hat{\theta}$ is thus such that $P(Y|x; \hat{\theta}) = P^*(Y|x)$ for all $x$ where $p_x(x) > 0$. Hence the same arguments after equation 7 in the main text lead the parametric equivalent to equation 3.1, which we summarize in the following corollary.

**Corollary A.2.** *If the true conditional $P^*(Y|x)$ is part of the parametric family $\{P(Y|x; \theta)\}_\theta$, then*

$$h_{\hat{\theta}}^i(x) - h_{\hat{\theta}}^j(x) = \log \frac{p_i(x)}{p_j(x)} \tag{15}$$

*for all $x$ where $p_x(x) = \sum_c \pi_c p_c(x) > 0$.*

We next derive conditions under which $\hat{\theta}$ is the unique minimum, which is needed to prove consistency. For that purpose, we perform a second-order Taylor expansion of $\mathcal{L}(\theta)$ around $\hat{\theta}$.

**Lemma A.3.**

$$\mathcal{L}(\hat{\theta} + \epsilon\phi) = \mathcal{L}(\theta) + \frac{\epsilon^2}{2} \phi^\top I \phi \tag{16}$$

*where $\epsilon > 0$ and $I = -\mathbb{E}_{p_x(x)} \mathbb{E}_{P^*(Y|x)}[H(Y, x)]$. The matrix $H(Y, x)$ contains the second derivatives of the log-model, i.e. its $(i, j)$-th element is*

$$[H(Y, x)]_{ij} = \frac{\partial^2}{\partial \theta_i \partial \theta_j} \log P(Y|x; \theta) \Big|_{\theta = \hat{\theta}} \tag{17}$$

*where $\theta_i$ and $\theta_j$ are the $i$-th and $j$-th element of $\theta$, respectively.*

*Proof.* A second-order Taylor expansion around $\mathcal{L}(\hat{\theta})$ gives

$$\mathcal{L}(\hat{\theta} + \epsilon\phi) = - \mathbb{E}_{p_x(x)} \sum_c P^*(Y = c|x) \log P(Y = c|x, \hat{\theta} + \epsilon\phi) \tag{18}$$

$$= \mathcal{L}(\hat{\theta}) - \nabla_\theta \mathcal{L}(\theta) \Big|_{\theta = \hat{\theta}} - \mathbb{E}_{p_x(x)} \sum_c P^*(Y = c|x) \frac{\epsilon^2}{2} \phi^\top H(Y = c, x)\phi + O(\epsilon^2) \tag{19}$$

$$= \mathcal{L}(\hat{\theta}) - \frac{\epsilon^2}{2} \phi^\top \left[ \mathbb{E}_{p_x(x)} \sum_c P^*(Y = c|x) H(Y = c, x) \right] \phi + O(\epsilon^2) \tag{20}$$

where we have used that the gradient of $\mathcal{L}(\theta)$ is zero at a minimizer $\hat{\theta}$. Since $\sum_c P^*(Y = c|x) H(Y = c, x) = \mathbb{E}_{P^*(Y|x)} H(Y, x)$, the result follows. $\qquad\square$

Note that $I(x) = -\mathbb{E}_{P^*(Y|x)}[H(Y, x)]$ is the conditional Fisher information matrix, and $I = \mathbb{E}_{p_x(x)}I(x)$ is its expected value taken with respect to $p_x(x)$.

**Corollary A.4.** *If $I$ is positive definite, then $\hat{\theta}$ is the unique minimizer of $\mathcal{L}(\theta)$.*

*Proof.* If $I$ is positive definite, then $\phi^\top I \phi > 0$ for all non-zero $\phi$ and by Lemma A.3, $\mathcal{L}(\hat{\theta} + \epsilon\phi) > \mathcal{L}(\hat{\theta})$ whenever $\phi \neq 0$. $\square$

We now consider the objective function $\mathcal{L}_n(\theta)$ where the expectations in $\mathcal{L}(\theta)$ are replaced by a sample average over $n$ samples. Let $\hat{\theta}_n = \arg\min_\theta \mathcal{L}_n(\theta)$.

**Proposition A.5.** *If (i) $I$ is positive definite and (ii) $\sup_\theta |\mathcal{L}_n(\theta) - \mathcal{L}(\theta)| \xrightarrow{p} 0$, then $\hat{\theta}_n \xrightarrow{p} \hat{\theta}$.*

*Proof.* By Corollary A.4, condition (i) ensures that $\hat{\theta} = \arg\min_\theta \mathcal{L}(\theta)$ is a unique minimizer, and hence that changing $\hat{\theta}$ by a small amount will increase the cost function $\mathcal{L}(\theta)$. Together with the technical condition (ii) on the uniform convergence of $\mathcal{L}_n(\theta)$ to $\mathcal{L}(\theta)$, this allows one to prove that $\hat{\theta}_n$ converges in probability to $\hat{\theta}$ as the sample size $n$ increases, following exactly the same reasoning as e.g. in proofs for consistency of maximum likelihood estimation (Wasserman, 2004, Section 9.13) or noise-contrastive estimation (Gutmann & Hyvärinen, 2012, Appendix A.3.2). $\square$

**Corollary A.6.** *If (i) $I$ is positive definite, (ii) $\sup_\theta |\mathcal{L}_n(\theta) - \mathcal{L}(\theta)| \xrightarrow{p} 0$, and (iii) there is a parameter value $\theta^*$ such that $P^*(Y|x) = p(Y|x; \theta^*)$, then $\hat{\theta}_n \xrightarrow{p} \theta^*$*

*Proof.* With Proposition A.5, condition (i) and (ii) ensure that $\hat{\theta}_n$ converges to $\hat{\theta} = \arg\min_\theta \mathcal{L}(\theta)$. With Lemma A.1, $\hat{\theta}$ is also minimizing $\mathbb{E}_{p_x(x)}\text{KL}\left(P^*(Y|x) \,||\, P(Y|x; \theta)\right)$. Hence, if condition (iii) holds, $\hat{\theta} = \theta^*$, and the result follows. $\square$

**Proposition A.7** (Consistency of the ratio estimator)**.** *If (i) $I$ is positive definite, (ii) $\sup_\theta |\mathcal{L}_n(\theta) - \mathcal{L}(\theta)| \xrightarrow{p} 0$, (iii) $P^*(Y|x) = p(Y|x; \theta^*)$ for some parameter value $\theta^*$, and (iv) the mapping from $\theta$ to $h_\theta^c$ is continuous, then*

$$h_{\hat{\theta}_n}^i(x) - h_{\hat{\theta}_n}^j(x) \xrightarrow{p} \log \frac{p_i(x)}{p_j(x)} \tag{21}$$

*for all $x$ where $p_x(x) = \sum_c \pi_c p_c(x) > 0$.*

*Proof.* By Proposition A.5, condition (i) and (ii) ensure that $\hat{\theta}_n$ converges to $\hat{\theta} = \arg\min_\theta \mathcal{L}(\theta)$. By Corollary A.2, condition (iii) ensures that $h_{\hat{\theta}}^i(x) - h_{\hat{\theta}}^j(x) = \log \frac{p_i(x)}{p_j(x)}$ for all $x$ where $p_x(x) = \sum_c \pi_c p_c(x) > 0$. Since continuous functions are closed under addition, the mapping from $\theta$ to $h_{\hat{\theta}}^i(x) - h_{\hat{\theta}}^j(x)$ is continuous if condition (iv) holds. We can then apply the continuous mapping theorem to conclude that $h_{\hat{\theta}_n}^i(x) - h_{\hat{\theta}_n}^j(x) \xrightarrow{p} h_{\hat{\theta}}^i(x) - h_{\hat{\theta}}^j(x) = \log \frac{p_i(x)}{p_j(x)}$, which establishes the result. $\square$

## B  Constructing $M$

We here elaborate on the three types of auxiliary distributions that we used in this work.

**Overlapping Distribution:**   The MDRE estimator, $\log \frac{p}{q} = \log \frac{p}{m} - \log \frac{m}{q}$ is defined when $p << m$ and $q << m$. Therefore, $m$ needs to be such that its support contains the supports of $p$ and $q$. Any distribution with full support such as the normal distribution trivially satisfies this requirement. However, satisfying this requirement does not guarantee empirical overlap of the distributions $p$, $q$ with $m$ in finite sample setting. In order to ensure overlap of samples between the two pairs of distributions we recommend the following:

- Heavy-tailed Distributions: Distributions such as, Cauchy and Student-t are better choice for $M$ compared to the normal distribution. This is because their heavier tails allow for easily connecting $p$ and $q$ with higher sample overlap when they are far apart (especially in the case of FOD).

- Mixtures: Another way to connect $p$ and $q$ using $m$ such that they have their samples overlap, is to use the mixture distribution. Here, we first convolve $p$ and $q$ with a standard normal and then take equal mixtures of the two.

- Truncated Normal: If $p$ and $q$ have finite support, one can also use a truncated normal distribution or a uniform distribution that at least spans over the entire support of $q$. This is assuming that $p << q$.

**Linear Mixing:** In this construction scheme, distribution $M$ is defined as the empirical distribution of the samples constructed by linearly combining samples $X_p = \{x_p^i\}_{i=1}^N$ and $X_q = \{x_q^i\}_{i=1}^N$ from distributions $p$ and $q$ respectively. That is, $m$ is the empirical distribution over the set $X_m = \{x_m^i | x_m^i = \alpha x_p^i + (1-\alpha) x_q^i, x_p \in X_p, x_q^i \in X_q\}_{i=1}^N$, where $\alpha$ is not constrained to create a convex mixture. This construction is related to the linear combination auxiliary of Rhodes et al. (2020). In TRE, the auxiliary distribution is defined as the empirical distribution of the set $X_m = \{x_m^i | x_m^i = \sqrt{1 - \alpha^2} x_p^i + \alpha x_q^i, x_p \in X_p, x_q^i \in X_q\}_{i=1}^N$, where $0 \leq \alpha \leq 1$. This weighting scheme skews the samples from the auxiliary distribution towards $p$. Therefore, care needs to be taken when $p$ and $q$ are finite support distributions so that the samples from the auxiliary distributions do not fall out of the support of $q$.

Using either of the weighting schemes, one can construct $K$ different auxiliary distributions. MDRE can either use these $K$ auxiliary distributions separately using a K+2-way classifier or define a single mixture distribution using them as component distributions and train a 3-way classifier. We refer to this construction as Mixture of Linear Mixing.

**Dimension-wise Mixing:** In this construction scheme, that is borrowed from TRE as it is, $M$ is defined as the empirical distribution of the samples generated by combining different subsets of dimensions from samples from $p$ and $q$. We describe the exact construction scheme from TRE below for completeness:

Given a $d$-length vector $x$ and that $d$ is divisible by $l$, we can write down $x = (x[1], ...x[l])$, where each $x[i]$ has length $d/l$. Then, a sample from the $k$th auxiliary distribution is given by: $x_k^i = (x_q^i[1], ...x_q^i[j], x_p^i[j + 1], ..., x_p^i[l])$, for $j = 1, ..., l)$, where $x_p^i \sim p$ and $x_q^i \sim q$ are randomly paired.

## C  1D density ratio estimation task

In Section 4.1, we studied three cases in which the two distributions $p$ and $q$ are separated by both FOD and HOD. In all of these 1D experiments, all models were trained with 100,000 samples, and all results are reported across 3 runs with different random seeds. Additionally, we found that MDRE worked equally well for 1K and 10K samples. The experimental configurations, including the auxiliary distributions for MDRE, are detailed in Table 5.

| $p$ | $q$ | True KL | MDRE @ 1K | MDRE @ 10K | MDRE @ 100K |
|---|---|---|---|---|---|
| $\mathcal{N}$(-1, 0.08) | $\mathcal{N}$(2, 0.15) | 200.27 | 195.05 | 196.50 | 203.32 |
| $\mathcal{N}$(-2, 0.08) | $\mathcal{N}$(2, 0.15) | 355.82 | 346.92 | 348.97 | 360.35 |

Table 4: MDRE on 1D density ratio estimation for three settings of sample sizes. MDRE estimates the density ratio well for all the three settings.

| $p$ | $q$ | TRE $p_k$ | MDRE $m$ |
|---|---|---|---|
| $\mathcal{N}(-1, 0.08)$ | $\mathcal{N}(2, 0.15)$ | Linear Mixing with $\alpha = [0.053, 0.11, 0.16, 0.21, 0.26, 0.31, 0.37, 0.42, 0.47, 0.53, 0.58, 0.63, 0.68, 0.74, 0.79, 0.84, 0.89, 0.95]$ | $\mathcal{C}(0, 1)$ |
| $\mathcal{N}(-2, 0.08)$ | $\mathcal{N}(2, 0.15)$ | Linear Mixing with $\alpha = [0.03, 0.07, 0.1, 0.14, 0.17, 0.21, 0.24, 0.28, 0.31, 0.34, 0.38, 0.41, 0.45, 0.48, 0.52, 0.55, 0.59, 0.62, 0.66, 0.69, 0.72, 0.76, 0.79, 0.83, 0.86, 0.9, 0.93, 0.97]$ | $\mathcal{C}(0, 1)$ |
| $\mathcal{N}(-10, 1)$ | $\mathcal{N}(10, 1)$ | Linear Mixing with $\alpha = [0.11, 0.22, 0.33, 0.44, 0.55, 0.66, 0.77, 0.88]$ | $\mathcal{C}(0, 2)$ |

Table 5: Experiment configurations for Table 1 of main text and Table 4 of Appendix C

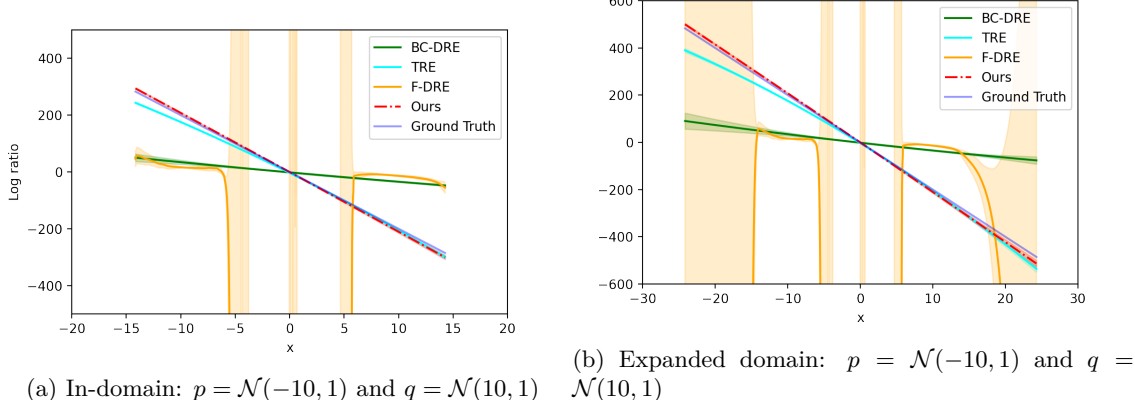

(a) In-domain: $p = \mathcal{N}(-10, 1)$ and $q = \mathcal{N}(10, 1)$

(b) Expanded domain: $p = \mathcal{N}(-10, 1)$ and $q = \mathcal{N}(10, 1)$

Figure 8: 1D density ratio estimation analysis. Figure 8a evaluates the log density ratios on uniform samples inside the domain of the respective training distribution. Figure 8b evaluates the log density ratios on uniform samples from an expanded domain of the training distribution. The shading represents 1 standard deviation of the estimates.

# D  Uncertainty Quantification of MDRE Log-ratio Estimates with Hamiltonian Monte Carlo

In the 1D experiments, MDRE consistently led to highly accurate KL divergence estimates even in challenging settings where state-of-the-art methods fail. To understand why MDRE gives such accurate KL estimates, we conduct an analysis on the reliability of its log-ratio estimates by analyzing the distribution of the estimates in a Bayesian setup, and study how it impacts the KL divergence estimation. For this analysis, we use a classifier with the standard normal distribution as the prior on its parameters. The distribution of the log-ratio estimates is simply the distribution of the estimates from the classifiers with different

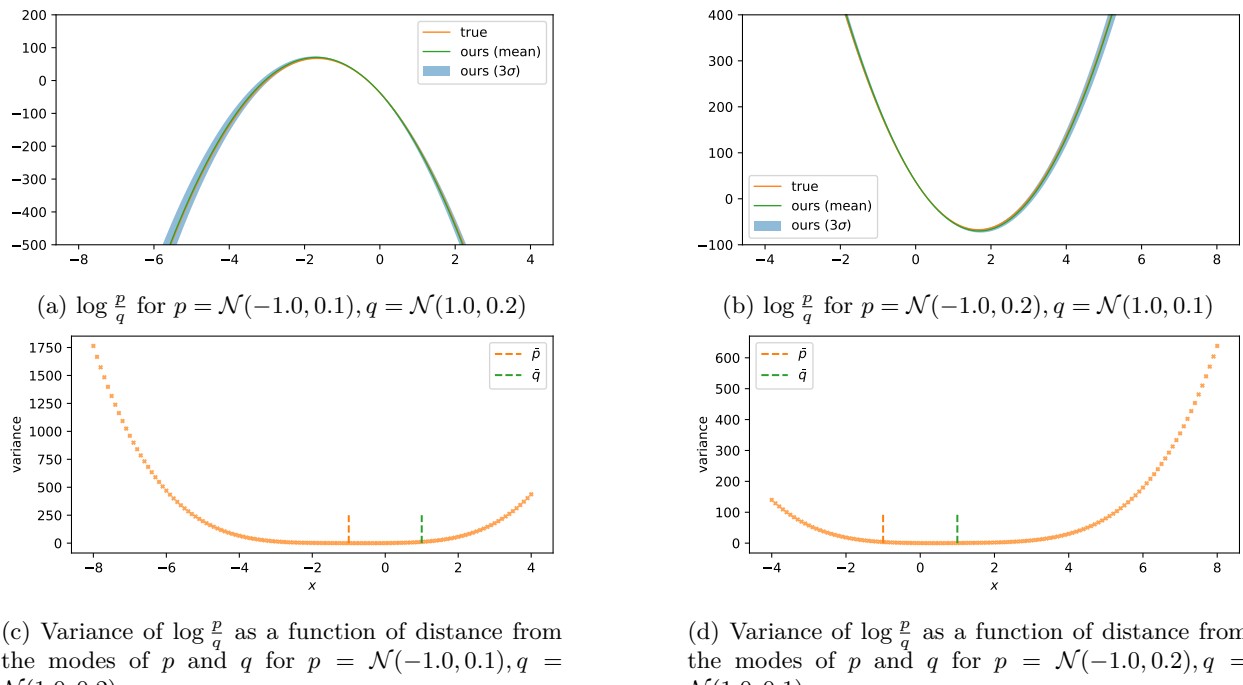

(a) $\log \frac{p}{q}$ for $p = \mathcal{N}(-1.0, 0.1), q = \mathcal{N}(1.0, 0.2)$

(b) $\log \frac{p}{q}$ for $p = \mathcal{N}(-1.0, 0.2), q = \mathcal{N}(1.0, 0.1)$

(c) Variance of $\log \frac{p}{q}$ as a function of distance from the modes of $p$ and $q$ for $p = \mathcal{N}(-1.0, 0.1), q = \mathcal{N}(1.0, 0.2)$

(d) Variance of $\log \frac{p}{q}$ as a function of distance from the modes of $p$ and $q$ for $p = \mathcal{N}(-1.0, 0.2), q = \mathcal{N}(1.0, 0.1)$

Figure 9: Uncertainty quantification for MDRE estimator. We plot the 3x standard deviation around the mean in light blue. In plots (c) and (d) the bars show the means of $p$ and $q$.

posterior parameters, which are sampled. We consider two setups where first we set $p = \mathcal{N}(-1.0, 0.1)$ and $q = \mathcal{N}(1.0, 0.2)$ and then swap their scales, i.e. $p = \mathcal{N}(-1.0, 0.1)$ and $q = \mathcal{N}(1.0, 0.2)$. In both the cases, we draw samples from the posterior using an Hamiltonian Monte Carlo (HMC) sampler initialized by the maximum likelihood estimate of the classifier parameter. We then compute a set of samples of the log-ratio estimates from MDRE and estimate the mean and standard deviation using these samples. Figure 9 (a) and (b) shows these results. We find that MDRE is accurate and manifests lowest uncertainty around the region between the means of $p$ ($-1.0$) and $q$ ($+1.0$). The uncertainty increases as we move away from the modes of distributions $p$ and $q$. This is shown in plots (c) and (d), where we plot the variance of the estimates as a function of the location of the sample.

Since KL divergence is the expectation of the log-ratio on samples from $p$ and the high density region of $p$ exactly matches the high confidence region of MDRE, it is able to consistently estimate the KL divergence accurately even when $p$ and $q$ are far apart.

# E  High Dimensional Experiment

In Section 4.2, we showed that MDRE performs better than all baseline models when $p$ and $q$ are high dimensional Gaussian distributions. Prior work of Rhodes et al. (2020) has considered high dimensional cases with HOD only, whereas we additionally consider cases with FOD and HOD to provide a more complete picture. Our results show that MDRE outperforms all other methods on the task of MI estimation as the function of the estimated density ratio. It is worth noting that MDRE uses only upto 5 auxiliary distributions that are constructed using the linear mixing scheme and beats TRE substantially on cases with both FOD and HOD, although TRE uses upto 15 auxiliary distributions also constructed using linear mixing approach. This demonstrates that our proposal of using the multi-class logistic regression does, in fact, prevent distribution shifts issues of TRE when both FOD and HOD are present and help estimate the density ratio more accurately.

We now describe the MDRE configuration and other setup related details.

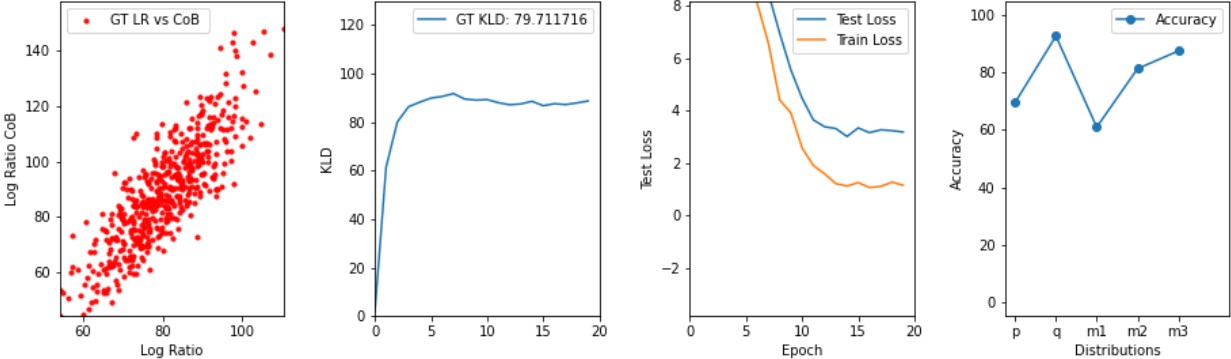

Figure 10: Diagnostic plot for a high dimensional experiment.

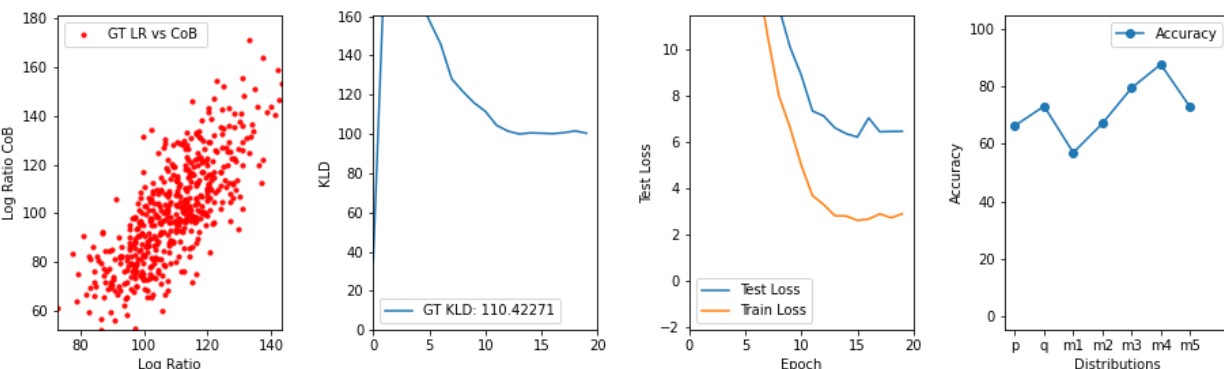

Figure 11: Diagnostic plot for a high dimensional experiment with randomized means.

| Dim | MI | $m$ **using LM** |
|---|---|---|
| 40 | 20 | [0.25,0.5,.75] |
| | 100 | [0.35,0.5,.85] |
| 160 | 40 | [0.25,0.5,.75] |
| | 136 | [0.15,0.35,0.5,.75,.95] |
| 320 | 80 | [0.25,0.5,.75] |
| | 240 | [0.15,0.35,0.5,.75,.95] |

Table 6: Configuration of MDRE for the high dimensional experiments. LM stands for Linear Mixing

**Auxiliary Distributions:** For all the high dimensional experiments throughout this work, we construct $m$ using the linear mixing scheme as described in Appendix B. Table 6 provides the number $K$ of auxiliary distributions along with the exact mixing weights for each of the 6 settings.

As a general principle, we chose these three sets of mixing weights so that their cumulative samples overlap with the samples of $p$ and $q$ similar to how the heavy tailed distribution worked in the 1D case. Please note that while heavy tailed distributions can effectively bridge $p$ and $q$ when they have high FOD. However, they do not work as well if the discrepancy is primarily HOD. For example consider $p = \mathcal{N}(0, 1e-6)$ and $q = \mathcal{N}(0,1)$. In this case, setting $m$ to a heavier tailed distribution centered as zero will not be of help. We need $m$ that is concentrated at zero but also maintains a decent overlap with $q$. Linear mixing $p$ and $q$ on the other hand, mixes first and higher order statistics (second or higher) and therefore, populates samples that overlap with both $p$ and $q$. In some cases, we found that a mixture of linear mixing with $K = 1$ can also be used to estimate the density ratio. However, this requires using a neural network-based classifier and requires much more tuning of the hyperparameters.

For choosing $K$, we use a grid search based approach. We monitor the classification accuracy across all the $K + 2$ distribution. If this accuracy is very high ($> 95\%$ for all classes), this implies that the classification task is easy and therefore the DRE may suffer from the density chasm issue. On the other hand, if the classification accuracy is too low ($< 50\%$ for all classes), then again, the DRE does not estimate well. We found that targeting an accuracy curve as shown in Figure 10 (last panel) empirically leads to accurate density ratio estimation. This curve plots the test accuracy across all the classes and, empirically when it stays between the low and the high bounds of (50%,95%), the DRE estimates the ratios fairly well. The first panel shows that MDRE estimates the ground truth ratio accurately across samples from all the $K + 2$ distributions, the second panel shows that KL estimates of MDRE is close to the ground truth KL and the third panel shows that both test and training losses have converged. Figure 11 shows another example for the case of randomized means. While MDRE also manages to get the ground truth KL correctly and most of the ratio estimates are also accurate, it does, however, slightly overestimate the log ratio for some of the samples from $p$.

# F  SpatialMultiOmniglot Experiment

SpatialMultiOmniglot is a dataset of paired images $u$ and $v$, where $u$ is a $n \times n$ grid of Omniglot characters from different Omniglot alphabets and $v$ is a $n \times n$ grid containing the next characters of the corresponding characters in $u$. In this setup, we treat each grid of $n \times n$ as a collection of $n^2$ categorical random variables, not the individual pixels. The mutual information $I(u,v)$ can be computed as: $I(u,v) = \sum_{i=1}^{n^2} \log l_i$, where $l_i$ is the alphabet size for the $i^{th}$ character in $u$. This problem allows us to easily control the complexity of the task since increasing $n$ increases the mutual information.

For the model, as in TRE, we use a separable architecture commonly used in MI-based representation learning literature and model the unnormalized log-scores with functions of the form $g(u)^T W f(u)$, where $g$ and $f$ are 14-layer convolutional ResNets (He et al., 2015). We construct the auxiliary distributions via dimension-wise mixing—exactly the way that TRE does.

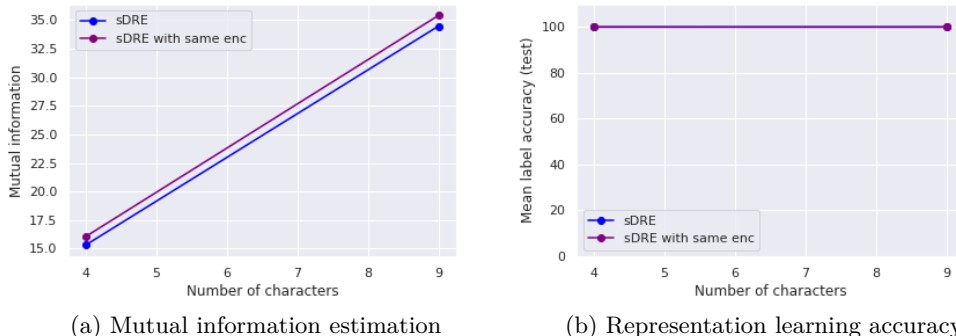

(a) Mutual information estimation      (b) Representation learning accuracy

Figure 12: SpatialMultiOmniglot representation learning results with same encoder for $f$ and $g$.

To evaluate the representations after learning, we adopt a standard linear evaluation protocol to train a linear classifier on the output of the frozen encoder $g(u)$ to predict the alphabetic index of each character in the grid $u$.

**Additional Experiments**

In addition to the experiments in the main text, we run an additional experiment with SpatialMultiOmniglot to test the effect of using the same encoder for $g$ and $f$ (i.e, modeling the unnormalized log-scores with the form $g(u)^T W g(v)$) instead of $\log p(u, v) = g(u)^T W f(v)$).

**Single Encoder Design:** We test the contribution of using two different encoders $f$ and $g$ instead of one. As seen in Figure 12, in both cases of $d = 4, 9$, the two models reach slightly different but similar MI estimates, but, interestingly, do not differ at all in the test classification accuracy. Empirically, we also found that using one encoder helps the model converge to much faster. Overall, this experiment demonstrates that using two different encoders does not necessarily work to our advantage.

# G   TRE on Finite Support Distributions for $K = 1$

For $K = 1$, TRE proposes the following telescoping: $\log p/q = \log p/m + \log m/q$. As such, for TRE to be well defined, $\frac{dM}{dQ}$ i.e. the Radon-Nikodym Derivatives (RND) needs to exist. The consequence of this is that TRE is only defined when $p << m << q$. However this condition easily breaks if, for example, $p$ and $q$ are mixtures of finite support distributions except for the trivial case when support of $m$ is exactly equal to the support of $q$.

We now demonstrate this with a specific example in Figure 13. Here we set $p = 0.5 \times \mathcal{TN}(-1, 0.1, low = -1.1, high = -0.9) + 0.5 \times \mathcal{TN}(1, 0.1, low = 0.9, high = 1.1)$ and $q = 0.5 \times \mathcal{TN}(-1, 0.2, low = -1.2, high = 0.8) + 0.5 \times \mathcal{TN}(1, 0.2, low = 0.8, high = 1)$, as shown in Figure 13a where $\mathcal{TN}$ stands for Truncated Normal distribution. We set the auxiliary distribution $m$ in TRE to $m = \mathcal{TN}(0, 1, low = -1.2, high = 1.2)$ using the proposed *overlapping distribution* construction. As such, $p << q << m$ and therefore, TRE is undefined for the second term as $\frac{m}{q}$ is not defined for samples from $m$ that are outside the support of $q$. It can be clearly seen in Figure 13b that $\frac{m}{q}$ blows up to very high values on samples from $m$ where $q$ does not have any support. Similar examples can be constructed for all the auxiliary distribution construction schemes proposed in Rhodes et al. (2020).

Please note, despite $dM/dQ$ being undefined, when used with the proposed $m$, TRE estimates $\log r_{p/q}$ accurately on samples from $p$. We conjecture that this is because, the numerical estimation of the $dM/dQ$ is finite over the support of $p$.

(a) Setup                                    (b) $dM/dQ$

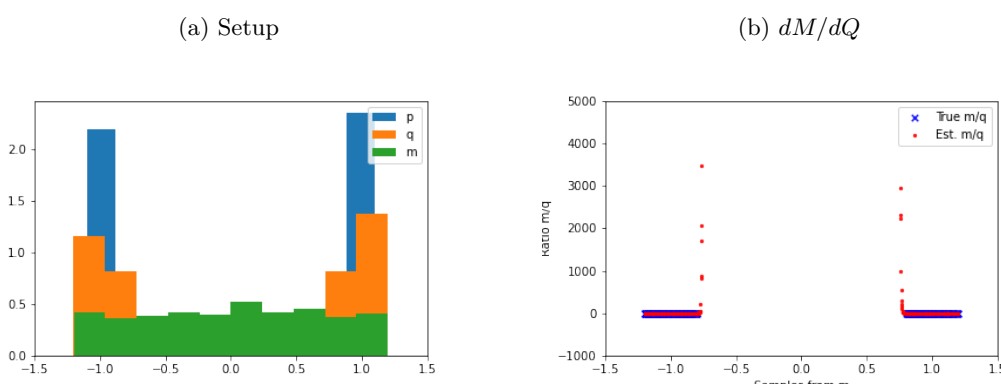

Figure 13: $dM/dQ$ on mixtures of distributions with finite support

