# OpenReview forum: "Estimating the Density Ratio between Distributions with High Discrepancy using Multinomial Logistic Regression"
_TMLR — Accepted by TMLR_

### Review · Reviewer_yAq2 · 2022-12-03

**Summary Of Contributions:**

The proposed work addresses the problem of estimating density ratios, which is relevant for a variety of tasks in machine learning and statistics, most notably the estimation of f-divergences.
A common approach is to use the output of a logistic regression classifier. However, these are known to perform poorly on well-separated distributions with extreme density ratios. In order to overcome this "density chasm", prior work has introduced (TRE) that uses intermediate distributions $m_{i}$ that allow an estimation as $\frac{p}{q} = \frac{p}{m_1}\frac{m_1}{m_2} \dots \frac{m_{K}}{q}$. Here, the intermediate distributions $\{m_i \}_{1 \leq i \leq K}$ are constructed to serve as a bridge from $p$ to $q$, with consecutive distributions strongly overlapping.

The proposed work observes that TRE's performance suffers from the "distribution shift" due to the fact that the density estimation problem for $\frac{m_{i}}{m_{i + 1}}$ is trained on data from $m_{i}$ and $m_{i + 1}$ but evaluated on data from only $p$ and $q$. To overcome this problem, the authors propose to obtain estimates for the density ratio from the $K+2$-class classification problem involving samples from all the $m_{i}$ and $p,q$.

**Audience:**

Yes

**Broader Impact Concerns:**

No concerns

**Claims And Evidence:**

Yes

**Requested Changes:**

In my opinion, the paper could be published as is.

**Strengths And Weaknesses:**

The paper makes a compelling case for a flaw in an existing method and proposes an elegant approach to fix it, with good empirical results. I don't really have anything to complain about.

---

### Review · Reviewer_775Q · 2022-12-07

**Summary Of Contributions:**

The paper tackles the density-chasm problem in density ratio estimation via logistic regression. The paper first shows that the existing approach to this problem, i.e., telescopic density-ratio estimation that uses the telescopic product of logistic regressors trained on overlapping auxiliary distributions to bridge the gap between target distributions, can be inaccurate, since a single inaccurate estimator in the product deteriorates the entire accuracy. The paper instead proposes to train a single multinomial logistic regressor on auxiliary distributions. The paper shows that in the infinite data regime this estimator is accurate and empirically estimates the estimator on a set of synthetic datasets as well as for MI estimation on the SpatialMultiOmniglot problem.

**Audience:**

Yes

**Claims And Evidence:**

Yes

**Requested Changes:**

- please show that the density-chasm problem has practical relevance, e.g., by showing how it can deteriorate representation earning.
- I think the paper would benefit greatly from proving that the proposed estimator is consistent.

**Strengths And Weaknesses:**

Strength:
- sound approach with proper theoretical underpinning
- proper empirical evaluation

Weakness:
- it is unclear how severe the density-chasm problem is in practice
- paper only hints at proving that the proposed estimator is consistent for finite samples

---

> ### Author Response · Authors · 2022-12-17
> **Manuscript Updated**
>
> We thank reviewer 775Q for the valuable feedback.
> ### Consistency Proof
> Per the reviewer's request, we have now added a new Appendix A with the consistency proof and amended Remark 3.3 accordingly.
>
> ### Practical relevance of the density-chasm problem:
> We demonstrate the impact of density-chasm on representation learning in section 4.3. In Figure 7, panel (a), we first show the mutual information (MI) as estimated using a BDRE, TRE, and MDRE. While MDRE estimates the ground-truth mutual information between the two grids of images, perfectly, BDRE and TRE do not. Then in panel (b), we show the accuracy of three classifiers that are trained upon the representations learned during the MI estimation task in panel (a) from BDRE, TRE, and MDRE. It can be seen that the test accuracy correlates very well with how accurately the three methods estimated the ground-truth MI. Thus, showing that density-chasm contributes to the deterioration of the quality of the learned representation.
>
> Similarly, Rhodes et al (2020) (Section 4.4) demonstrated that unlike BDRE, which suffers from the density-chasm problem more severely, TRE leads to better visual fidelity and likelihood estimates in energy models as a virtue of mitigating the density-chasm problem.
>
> Other practical use cases: Mutual information estimation, Covariate shift, and Importance sampling. We will update the manuscript to enumerate these use cases.
>
> [1] Rhodes, Benjamin, Kai Xu, and Michael U. Gutmann. "Telescoping density-ratio estimation." Advances in neural information processing systems 33 (2020): 4905-4916.

---

### Review · Reviewer_58JZ · 2022-12-30

**Summary Of Contributions:**

The paper "Estimating the Density Ratio between Distributions with High Discrepancy using Multinomial Logistic Regression" proposes to deal with a well-known problem, called the chasm problem, for density ratio estimations, which arises when the two compared distributions have two strongly disjoint supports. The paper starts by pointing limitations of related work dealing with this problem, namely the distribution shift from which they suffer, and then demonstrates that the simple use of a cross-entropy loss for the discrimination of a set of auxiliary distributions (including source and target distributions) enable to obtain clearly better calibrated results than previous work.

**Audience:**

Yes

**Broader Impact Concerns:**

No concern about ethical implications of the work to be mentioned.

**Claims And Evidence:**

Yes

**Requested Changes:**

While I found the paper interesting and results promising for many applications, I am not convinced by the theoretical part. Since obviously $p_i(x)=p(x|y=i)=\frac{p(x)p(y=i|x)}{\pi_i}$, if we set an estimator $\hat{p}(y=i|x) = \frac{\pi_i exp(h_i(x))}{\sum_k \pi_k exp(h_k(x))}$, we have $\hat{p}_i(x)=\frac{p(x) exp(h_i(x))}{\sum_k \pi_k exp(h_k(x))}$ and thus  $log \frac{\hat{p}_i(x)}{\hat{p}_j(x)}=log \frac{exp(h_i(x))}{exp(h_j(x))} = h_i(x) - h_j(x)$. Hence,  $h_i(x) - h_j(x)$ is an estimator of the log density-ratio. This justifies the use of this quantity to estimate the ratio in the approach. However, the derivation with optimizers  $\hat{h}_i(x)$ to get (6) and (7) does not look correct to me, because this should not be taken for individual x samples but in expectation: optimization is performed on (5) with expectations leading to $\frac{\partial L}{\partial h_i}= - \pi_i + \int p_x(x) \frac{\pi_i exp(h_i(x))}{\sum \pi_k exp(h_k(x))}$ which does not allow to conclude in (7) for all x when cancelling it. Could authors elaborate on this ? And also give more insights about why this derivation is needed (rather than simply giving obvious equations that I give at the beginning of this paragraph) ?

More importantly, while it could explain the absence of distribution shift, it gives no insights about why using auxiliary distributions in that case would allow to deal with the chasm problem. While this is quite intuitive for TRE from its construction based on a product of auxiliary ratios, for me the reason why the MDRE proposal obtains interesting results lies in the fact that all h share parameters which induces some regularization allowing better estimation. Please discuss further on the benefits of the approach over a classical BDRE approach.

Another point that I would be happy to see further discuss is the impact of the choice of the used auxiliary distributions on the results. Is it enough to only cover the full support of p and q? Is a single auxiliary enough or is it better to consider multiple distributions ? Does it have a correlation with the freedom of the network h ?

As the paper is very related to the calibration field, It would be useful to further discuss the positioning w.r.t other approaches for this task, which go far beyond the use of auxiliary distributions.

I also feel it would be useful to discuss perspectives of applications  of that work in domains considering complex data structures as sequences. For instance, density ratios are of crucial importance in imitation learning which induces sequential data. Would it be applicable in such domains ?

Minor remarks:
   -Although it does not change anything in the following, (6) should be multiplied by (-1) to correspond to a derivative.
   - Please indicate the axes of panels b and d of figure 1

**Strengths And Weaknesses:**

Strengths:
    - Very simple approach, very easy to reproduce
    - Impressive results compared to state of the art
    - Very important problem for many domains - density ratio estimation is at the heart of many research works in machine learning

Weakness:
    - Theoretical part not fully convincing (see next section)
    - Reasons of benefits not clearly identified / discussed
    - Positioning w.r.t. state of the art in calibration or density ratio estimation not sufficient from my point of view.
    - More insights about the impact of the choice of auxiliary distribution would be useful

---

> ### Author Response · Authors · 2023-01-09
> **Response**
>
> We thank the reviewer for their valuable feedback and comments!
>
> 1. Correctness of derivation of the estimator: In going from 5 to 6, as stated inline, we take the *functional* derivative of 5 wrt to $h_i(x)$, and hence the expectation term is not present in 6. The functional derivative is sometimes also called the variational derivative. An introduction can be found here: [https://www.youtube.com/watch?v=OB7xlRsKlKM](https://www.youtube.com/watch?v=OB7xlRsKlKM) or [https://mbernste.github.io/posts/functionals/](https://mbernste.github.io/posts/functionals/). While the functional is defined in terms of an integral, the functional derivative does not feature the integral, see around minute 18 of the video, Equation (2.5). Please further note that, as stated in the paper after equation (6), the equation holds “for all x where $p_x(x)>0$. This reflects the influence of the integral, which the reviewer was probably wondering about.
>
> 2. Simple Proof: While the reviewer’s derivation provides some motivation for our result, we are afraid, it is actually not a proof as the relation between the MDRE estimator and the learning objective is missing. Our proof shows how the softmax cross entropy loss connects to density ratio estimation, which allows us to study identifiability and consistency of our estimator. Hence, in a sense, we “set” our estimator to the target quantity, which the reviewer mentions, by minimizing the loss function. An additional, more rigorous, proof that does not require the functional derivative is provided in Appendix A of the updated manuscript.
>
> 3. Density Chasm: Density chasm happens when the classifier fails to find the decision boundary corresponding to the Bayes optimal classifier. This is common when the distributions are well-separated, making the classification problem ‘easy’, since there exist a number of decision boundaries that perfectly discriminate a finite amount of samples from the two distributions. We show this in Figure 1, panel (a) for BDRE. Both TRE and MDRE use overlapping auxiliary distributions (albeit a little differently) to mitigate this issue by making the classification problem(s) harder.
>
>     In MDRE, we achieve it in by first introducing auxiliary distribution(s) that overlap with the samples from the original two distributions, hence making their discrimination harder. Then we cast the original density ratio estimation problem as a multi-class classification task to estimate the ratio without suffering from distribution-shift issues. As shown in panel (c) of Figure 1, this allows for significantly improving the quality of the learned decision boundary, which in turn leads to accurate density ratio estimation.
>
>     While parameter sharing could certainly help, we would like to emphasise that except for the representation learning experiments, as stated in the experiment sections 4.1 and 4.2, there is no parameter sharing across the classes in MDRE architecture. We have now clarified this in the revised paper.
>
>     Moreover, a theoretical justification for how MDRE addresses the density chasm is given in Remark 3.5. The remarks shows that MDRE yields correct results for all inputs $x$ that are in the union of the domain of the distributions of interest, p and q, and the auxiliary distributions. This is in contrast to TRE where the learned ratios are only correct in the domain of the individual denominator distributions. This is explained in Section 2.1 in the paper.
>
> 4. Choice of Auxiliary Distribution: We agree that choice and number of auxiliary distributions have a significant impact on the overall performance of the MDRE estimator. But, this choice depends on the problem domain as well as its modality and dimensionality. As such, an exhaustive description of auxiliary measures is beyond the scope of the current work, leaving learning based approaches to the construction of auxiliary distributions, as an open research question. However, in this work we have provided a diverse set of empirical results and a few recipes for constructing auxiliary distributions. We provide our strategy for picking K for the high dimensional experiment in Appendix E. Moreover, for the case of representation learning task, we study the impact of changing the number of such distributions in Figure 7, panel (C).
>
> ... Continued on the next comment due to space limitation.

---

> > ### Author Response · Authors · 2023-01-09
> > **Continued...**
> >
> > 5. Other Approaches:  It is not exactly clear to us what the reviewer means with the "calibration field" and what other prior work they have in mind. If important to them, could they please clarify?
> > Assuming calibration refers to estimation of classifiers in supervised learning, then, yes, our approach is closely related since we frame the problem of density ratio estimation in terms of multinomial regression. In sections 1 and 2, we discuss further related work. Moreover, the calibration issue in classifiers has been studied for likelihood ratio testing (e.g. [1]). However, neither prior work nor the current exposition establishes a formal connection between density chasm and the calibration of the classifier. While this is an interesting avenue, we leave it as future work direction.
> > 6. Minor remarks: - Yes, the sign in Eq 6 should be flipped, thanks. - We will update the labels.
> >
> > [1] Cranmer, Kyle, Juan Pavez, and Gilles Louppe. 2015. “Approximating Likelihood Ratios with Calibrated Discriminative Classifiers.” *arXiv [stat.AP]*
> > . arXiv. http://arxiv.org/abs/1506.02169.

---

### Review · Reviewer_jg6m · 2022-12-31

**Summary Of Contributions:**

The paper studies the problem of density ratio estimation and, in particular, tackles the density-chasm problem.

The proposed approach falls under telescopic ratio estimation methods that construct a sequence of auxiliary distributions to bridge the two targeted densities. A prior workstream called BDRE converts the original ratio estimation task into binary classification problems and recovers the desired ratio by the telescopic product of the neighboring distributions' ratio estimates. But as the authors showed experimentally, the BDRE methods can deviate significantly sometimes under train-eval distribution shifts.

The paper addresses the deviation issue by reforming the regressor as a multi-nomial logistic regression problem with a log-likelihood objective. The new approach imposes implicit regularization onto the final estimator and makes the estimator less sensitive to distribution shifts. The following experimental results and some specific examples further demonstrate the advantage of the new method in consistency and accuracy.

**Audience:**

Yes

**Broader Impact Concerns:**

The reviewer has not identified any broader impact concerns.

**Claims And Evidence:**

No

**Requested Changes:**

- It would be great if the authors could address the abovementioned weaknesses.
- A suggestion is to add more insights on why resolving the "density-chasm problem" is practically essential.

**Strengths And Weaknesses:**

Strength:
 - The approach leverages some standard formulations and natural ideas and is easy to follow.
 - The experimental results showed significant differences between the performance of the new algorithm and existing approaches.
 - The proposed ratio estimator is consistent under some additional assumptions.

Weaknesses:
 - The main contribution of the new approach is addressing the distribution-shift issue. But Section 3.3, titled "Free from distribution-shift problems," mainly relies on some 1D examples. Hence there doesn't seem to be a concrete theoretical justification for the contribution.
 - The paper stated that the new method "has better finite-sample (non-asymptotic) behavior than current state-of-the-art methods." Is this referring to empirical performance? The reviewer has not found much theoretical support for the method being better than SOTA. Similarly, another statement said that "MDRE solves the aforementioned distribution shift issue." The word "solve" might be too strong here if the method only empirically addresses the issue in some examples.

---

> ### Author Response · Authors · 2023-01-09
> **Response**
>
> We thank the reviewer for their time to provide us with their valuable feedback!
>
> 1. Summary: We would like to clarify certain points made in the summary which may not accurately describe the current work.
>
>
>     > The proposed approach falls under telescopic ratio estimation methods
>     >
>
>     MDRE is not a telescoping based density ratio estimation approach. This is shown in Proposition 3.1 where instead of telescoping, we show that the ratio is simply the difference of the corresponding logits.
>
>     > A prior workstream called BDRE converts the original ratio estimation task into binary classification problems and recovers the desired ratio by the telescopic product
>     >
>
>     This is perhaps just a typo, we assume that the reviewer meant TRE instead of BDRE.
>
>
>     > The new approach imposes implicit regularization onto the final estimator and makes the estimator less sensitive to distribution shifts.
>     >
>
>     The mitigation of the distribution-shift problem is primarily due to the construction of MDRE and not regularization. More specifically, unlike TRE, MDRE estimator is trained and tested on the samples from the same set of distributions. Furthermore, while parameter sharing could certainly provide some implicit regularization, we would like to clarify that except for the representation learning experiments, as stated in the experiment sections 4.1 and 4.2, there is no parameter sharing across the classes in MDRE architecture.
>
>
> 2. Distribution Shift: MDRE resolves distribution-shift by construction in the sense of Remark 3.5. We agree that Section 3.3 focuses on the illustrative example and, while we point to Remark 3.5, this could be made more prominent. We thus changed the first sentence as follows:
>
>     > "We continue with the example task of estimating the density ratio between p = N (−1, 0.1) and q = N (1, 0.2) and here illustrate Remark 3.5 that MDRE does not suffer from the distribution shift problem identified in Section 2.1.”
>     >
>
>     Regarding the "solving the distribution issue", the reviewer likely refers to the sentence "MDRE solves the aforementioned distribution shift issue by construction and [...]". Here, the "by construction" is important since, as explained in Remark 3.5, it is the construction of the estimator that resolves the problem. We agree, however, that "solving" can appear too strong and changed the sentence to "MDRE re-solves the aforementioned distribution shift issue by construction and [...].”
>
>
> 3. Finite sample performance: We do not intend make theoretical claims about the finite-sample performance of MDRE; we establish this via an extensive set of empirical results in our work. In order to clarify this point further, we have amended the abstract and the following sentence in the discussion section:
>
>     > "In this work, we presented the multinomial logistic regression based density ratio estimator (MDRE), a new method for density ratio estimation that has better finite sample (non-asymptotic) behavior than current state-of-the-art methods.”
>     >
>
>     Is changed to:
>
>     > "In this work, we presented the multinomial logistic regression based density ratio estimator (MDRE), a new method for density ratio estimation that had better finite sample (non-asymptotic) performance in our simulations than current state-of-the-art methods.”
>     >
>
>     Theoretical claims are made regarding the asymptotic behaviour i.e. consistency, which we have added now in Appendix A.
>
>
> 4. Practical Relevance of Density-Chasm:  We demonstrate the impact of density-chasm on representation learning in section 4.3. In Figure 7, panel (a), we first show the mutual information (MI) as estimated using a BDRE, TRE, and MDRE. While MDRE estimates the ground-truth mutual information between the two grids of images perfectly, BDRE and TRE do not. Then in panel (b), we show the accuracy of three classifiers that are trained upon the representations learned during the MI estimation task in panel (a) from BDRE, TRE, and MDRE. It can be seen that the test accuracy correlates very well with how accurately the three methods estimated the ground-truth MI. Thus, our result show that addressing the density-chasm contribute to increasing the quality of the learned representation.
>
>     Similarly, Rhodes et al (2020) (Section 4.4) demonstrated that unlike BDRE, which suffers from the density-chasm problem more severely, TRE leads to better visual fidelity and likelihood estimates in energy models as a virtue of mitigating the density-chasm problem.
>
>     Other practical use cases are: Mutual information estimation, covariate shift, and importance sampling. We have updated the manuscript to enumerate these use cases.
>
> [1] Rhodes, Benjamin, Kai Xu, and Michael U. Gutmann. "Telescoping density-ratio estimation." Advances in neural information processing systems 33 (2020): 4905-4916.

---

### Decision · Action_Editors · 2023-03-07

**Recommendation:** Accept with minor revision

**Comment:**

In this paper, the authors present a novel way to estimate the density ratio under distribution shifts. The key idea is to introduce a set of $K$ auxiliary densities, and leverage multi-class classification for density ratio estimation. Specifically, they show that if these auxiliary densities overlap with distributions $p$ and $q$, then a multi-class logistic regression can estimate $\log p/q$ on the domain of any of the $K + 2$ distributions. Finally, the authors demonstrate superior performance of their method in the experiments.

The idea of using multi-class logistic regression to estimate the density ratio is novel and interesting. The algorithm is very simple and works well in the experiments. The authors have made a significant contribution to density ratio estimation, and all the reviewers are positive with this paper. When preparing the camera-ready version, the authors should take all the discussions into consideration.


**Audience:**

Yes, estimating the density ratio is an important problem in machine learning.

**Claims And Evidence:**

Yes